# LLM Meeting Decision Trees on Tabular Data

**Hangting Ye[1], Jinmeng Li[1], He Zhao[2 3], Dandan Guo[1 *], Yi Chang[1 4 5*]**

School of Artificial Intelligence, Jilin University[1]
CSIRO's Data61[2]; Monash University[3]; International Center of Future Science, Jilin University[4]
Engineering Research Center of Knowledge-Driven Human-Machine Intelligence, MOE, China[5]
{yeht2118,lijm9921}@mails.jlu.edu.cn, he.zhao@data61.csiro.au,
{guodandan,yichang}@jlu.edu.cn

## Abstract

Tabular data have been playing a vital role in diverse real-world fields, including healthcare, finance, etc. With the recent success of Large Language Models (LLMs), early explorations of extending LLMs to the domain of tabular data have been developed. Most of these LLM-based methods typically first serialize tabular data into natural language descriptions, and then tune LLMs or directly infer on these serialized data. However, these methods suffer from two key inherent issues: (i) data perspective: existing data serialization methods lack universal applicability for structured tabular data, and may pose privacy risks through direct textual exposure, and (ii) model perspective: LLM fine-tuning methods struggle with tabular data, and in-context learning scalability is bottle-necked by input length constraints (suitable for few-shot learning). This work explores a novel direction of integrating LLMs into tabular data through logical decision tree rules as intermediaries, proposing a decision tree enhancer with LLM-derived rule for tabular prediction, DeLTa. The proposed DeLTa avoids tabular data serialization, and can be applied to full data learning setting without LLM fine-tuning. Specifically, we leverage the reasoning ability of LLMs to redesign an improved rule given a set of decision tree rules. Furthermore, we provide a calibration method for original decision trees via new generated rule by LLM, which approximates the error correction vector to steer the original decision tree predictions in the direction of "errors" reducing. Finally, extensive experiments on diverse tabular benchmarks show that our method achieves state-of-the-art performance. The source code is available at https://github.com/HangtingYe/DeLTa.

## 1 Introduction

Tabular data, typically organized in a structured table format within a relational database with rows and columns standing for the data samples and heterogeneous features (e.g., categorical and numerical features), is fundamental in various real-world fields, including healthcare [1], advertising [2], finance [3], etc. Given the heterogeneity of features [4, 5, 6, 7], decision tree-based methods [8, 9, 10] were found to be particularly suitable for tabular data [11, 12, 13]. While deep learning has led to breakthroughs in computer vision [14] and natural language processing [15], decision tree-based methods still outperform the majority of existing deep tabular methods on tabular prediction tasks such as classification and regression. This superiority of decision tree-based methods is explained by the feature heterogeneity for tabular data, where neural networks struggle to learn the irregular target functions compared to decision tree-based methods [11, 16].

Recently, LLMs have exhibited remarkable capabilities in natural language understanding and reasoning [17, 18], sparking growing interest in applying LLMs to structured tabular data tasks [19,

---

*Corresponding authors.

39th Conference on Neural Information Processing Systems (NeurIPS 2025).

Table 1: Most existing LLM-based tabular prediction methods require access to tabular samples and either train the LLMs or perform inference via in-context learning. Many of them are restricted to constrained settings such as few-shot or classification-only settings. The applicability of each scenario is determined on the experimental setups in the original papers. For instance, if a method primarily focuses on for few-shot learning, we mark the few-shot setting as applicable (✔); if it can also be extended to full-data training, we additionally mark the full-data setting as applicable (✔).

| Methods | No LLM access sample required | No LLM training required | | Applied scenario | | | |
|---|---|---|---|---|---|---|---|
| | | No pre-training | No fine-tuning | Full data | Few shot | Classification | Regression |
| TabLLM (2023) [20] | ✗ | ✔ | ✗ | ✔ | ✔ | ✔ | ✗ |
| LIFT (2022) [19] | ✗ | ✔ | ✗ | ✔ | ✔ | ✔ | ✔ |
| TP-BERTa (2024) [21] | ✗ | ✗ | ✗ | ✔ | ✔ | ✔ | ✔ |
| GTL (2024) [22] | ✗ | ✗ | ✔ | ✗ | ✔ | ✔ | ✔ |
| SERSAL (2025) [25] | ✗ | ✔ | ✗ | ✔ (Unsupervised) | ✔ | ✔ (Binary classification) | ✗ |
| P2T (2024) [23] | ✗ | ✔ | ✔ | ✗ | ✔ | ✔ | ✔ |
| FeatLLM (2024) [24] | ✗ | ✔ | ✔ | ✔ | ✔ | ✔ | ✗ |
| **DeLTa (Ours)** | ✔ | ✔ | ✔ | ✔ | ✔ | ✔ | ✔ |

20, 21, 22, 23, 24, 25]. Typically, most of the existing methods first serialize tabular samples into natural language descriptions, and then tune LLMs or directly infer via in-context learning on these serialized data. Despite the success of LLMs on text-based tasks, leveraging LLMs to empower tabular predictions remains a challenging task. Through an in-depth investigation of prior LLM-based approaches for tabular data, we identify two inherent characteristics hindering prediction:

**(i)** *Data perspective***:** To bridge the modality gap between unstructured text and structured tabular data with heterogeneous features, most existing methods serialize tabular samples into text formats. For instance, TabLLM [20] converts a sample into listed feature descriptions such as "The [column name] is [feature value]", and demonstrates that this template facilitates very-few-shot classification by leveraging the semantic priors of column names and values already encoded in LLMs. However, many real-world tabular datasets anonymize feature names using placeholder symbols for privacy [22] (e.g., finance), reducing the effectiveness of serialization methods that rely on semantic feature descriptors. In addition, LLMs tend to be less sensitive to numerical features [26, 21]. The predefined templates with inserted values often yield text formats that make LLMs struggle to understand the inherent interactions among different features. Beyond this, directly exposing the serialized tabular samples containing raw feature values to LLMs raises significant privacy concerns, especially in sensitive domains such as healthcare and finance, where data security is crucial [27, 28]. These challenges make it difficult to design a universally suitable serialization template for tabular data.

**(ii)** *Model perspective***:** One straightforward way to adapt LLMs to tabular prediction tasks is fine-tuning LLMs' parameters. For example, LIFT [19] investigated the fine-tuned GPT-3 models [17] on tabular data, revealing that the performance of fine-tuned LLMs was roughly on par with traditional tabular prediction methods. Despite the effectiveness, fine-tuning LLMs remains a challenging task even with recent parameter-efficient fine-tuning methods [29], especially in structured tabular data [19]. Another research line employs in-context learning by adding few-shot example demonstrations to the prompts without training, which is evidenced by LLMs' impressive capacity to learn from demonstrations included as part of the prompt [17]. But most of these methods are inherently limited by the input length constraints of LLMs, restricting their use to few-shot, or classification-only tasks. Therefore, LLMs still struggle to make satisfactory predictions for tabular data. We summarize these related works in Table 1.

In this paper, we explore a novel direction of integrating LLMs into tabular data via logical decision tree rules [30] as intermediaries. We propose DeLTa, a decision tree enhancer with LLM-derived rule for tabular prediction. Specifically, we first leverage the reasoning ability of LLMs to redesign an improved rule given a decision tree rule set, enforcing greater coherence among samples falling into the same leaf node. The newly generated rule is used to approximate a sample-specific error correction vector to calibrate the prediction of original decision trees in the direction of "errors" reducing. The proposed DeLTa could well solve the aforementioned challenges: *Solving the data issue:* Unlike serialization methods that convert each sample into unnatural text formats, decision tree rules are composed of simple comparisons between feature values and thresholds, forming logical, interpretable structures that can be naturally expressed in text without relying on semantic columns names. In addition, decision tree rules represent global feature space partitioning rule rather than individual samples, which helps mitigate privacy concerns by avoiding exposure of sample-level

information. *Solving the model issue:* Moreover, the powerful reasoning ability of LLMs can be leveraged to redesign decision tree rules and help trees with aggregating their decisions, rather than directly using LLMs to generate label predictions. Notably, DeLTa avoids serialize tabular data into natural language format, and does not require additional domain-specific expertise or semantic information, such as explicit feature names and detailed task background knowledge. Furthermore, DeLTa can be applied in full data learning setting without LLM fine-tuning.

The contributions of this paper include: 1) We investigated the prior LLM-based methods for tabular prediction and explore a novel direction of integrating LLMs into tabular data via refining the decision tree rules, without directly accessing to the data itself, addressing the inherent issue w.r.t. data perspective and model perspective. 2) We propose a decision tree enhancer with LLM-derived rule for tabular prediction, DeLTa, which utilizes LLMs to refine a set of decision tree rules derived on decision trees trained on multiple train subsets, where the newly generated rule is used to infer the sample-wise error correction vector to calibrate the output of original decision trees. 3) We conduct extensive experiments on various tabular benchmarks and competing benchmark algorithms, and comprehensive results along with analysis and visualizations demonstrate our effectiveness.

## 2   Related work

**Machine learning for tabular prediction.** The development of effective algorithms for predictive modeling on tabular data has been a longstanding research topic. In the early days, decision tree-based methods (e.g. XGBoost [8], CatBoost [10]) were found to be particularly suitable for tabular data. More recently, inspired by the success of of deep learning in computer vision (CV) [14] and natural language processing (NLP) [15], numerous methods have been proposed for tabular data to accomplish tabular prediction tasks. These works mainly include MLP-like models [31, 5, 32, 33], attention-based architectures [34, 5, 35, 36, 37], and retrieval-augmented architectures [38, 35, 39, 40]. Among these works, DCN V2 [31] is an architecture that consists of an MLP-like module and a feature crossing module; AutoInt [34] leveraged the Transformer architecture to capture inter-column correlations; FT-Transformer [5] further enhanced AutoInt's performance through improved token embeddings; ModernNCA [38] makes predictions based on the relationships with neighbors in a learned embedding space. Recently, another line of research has tried to use additional information outside target dataset to enhance tabular data prediction. XTab [41] pretrains Transformer on a variety of datasets for cross-table pretraining. TabPFN [42, 43], which is pretrained on a large set of synthetic datasets, serves as a foundation model for small to medium-sized tabular data.

**Large language models for tabular prediction.** Motivated by the impressive success of LLMs [17, 18, 44, 45], another promising study has attempted to harness the rich prior knowledge encapsulated by LLMs in tabular prediction tasks. LIFT [19] investigated the performance of fine-tuned GPT-3 models [17] on tabular data, revealing that the performance of fine-tuned LLMs was roughly on par with traditional solutions. Extending this line of research, TabLLM [20] employed T0 [46] as the base LLM and demonstrated competitive performance of fine-tuned LLMs in very few-shot scenarios. Additionally, recent efforts have focused on pre-training LLMs over diverse tabular datasets from different domains. Among these, TP-BERTa [21] introduces a tabular-specific tokenization scheme, enabling a single pre-trained language model to generalize across multiple tabular datasets after further fine-tuning. GTL [22] further promotes comprehensive instruction-following capabilities for both zero-shot and in-context learning with a limited number of examples. Despite their promising results, these approaches typically require training or fine-tuning LLMs. Instead of directly fine-tuning LLMs, P2T [23] leverages unlabeled data correlated with target data expressed in natural language form and prompts LLMs for few-shot semi-supervised learning. Recently, Summary [47] proposed a boosting framework that treats LLMs as weak learners for tabular prediction, particularly for tasks involving small numbers of data points. FeatLLM [24] prompts LLMs with serialized training examples to generate new features for few-shot classification tasks. SERSAL [25] explores to use synergy learning with FT-Transformer to enhance the noisy annotations generated by LLMs for binary classification tasks in unsupervised manner. This method also needs to fine-tune the LLMs. Going beyond tabular prediction, Nam et al. [48] leverages LLMs to optimize the feature generator with tree rules in the field of feature engineering [49], which we do not consider as a close related work to ours as the primary goal, motivation, and methodology are different.

Although prior works demonstrate that LLMs can make predictions for tabular tasks, the majority of these approaches rely on first converting tabular data into natural language descriptions, i.e.,

serialization, which often produces unnatural texts that differ from how humans might describe the data [47]. And it is also challenging to design such suitable template. In addition, these methods suffer from two limitations that either need to train LLMs or are restricted to constrained settings such as few-shot or classification-only settings. Table 1 summarizes these related works. The above limitations raise a fundamental question: can we harness the capabilities of LLMs without fine-tuning and without serializing tabular data, to improve full-data tabular prediction tasks across both classification and regression?

## 3 Preliminaries

**Problem formulation.** Denote a tabular dataset $\mathcal{D} = \{(x_i, y_i)\}_{i=1}^N$ as a collection of $N$ samples, where $(x_i, y_i)$ is the $i$-th data pair with $x_i \in \mathbb{X}$ representing input features and $y_i \in \mathbb{Y}$ the corresponding label. Concretely, we have $x_i = (x_i^{(num)}, x_i^{(cat)})$ where $x_i^{(num)}$ and $x_i^{(cat)}$ represent the numerical and categorical features respectively. We consider supervised tabular prediction tasks: binary classification $\mathbb{Y} = \{0, 1\}$, multiclass classification $\mathbb{Y} = \{1, ..., c\}$ and regression $\mathbb{Y} = \mathbb{R}$. For data splits, $\mathcal{D}_{train}$ denotes training set for model training, $\mathcal{D}_{val}$ validation set for early stopping and hyperparameter tuning, and $\mathcal{D}_{test}$ test set for final evaluation. The goal is to obtain an accurate model $G : \mathbb{X} \to \mathbb{Y}$ trained on $\mathcal{D}_{train}$, that minimizes the expected loss $\mathbb{E}[\mathcal{L}(G(x), y)]$. Here, $\mathcal{L}$ is a smooth loss function (e.g., mean squared error or cross-entropy) and $G$ can be any tabular predictive model.

**Decision trees (DTs).** A decision tree (DT) [30] is a hierarchical model that recursively partitions the input feature space into disjoint regions, i.e., leaf nodes, through a series of axis-aligned splits. For a given partitioning rule $r$, the prediction function $f$ can be formally expressed as:

$$f(x|\mathcal{D}_{train}, r) = \sum_{l=1}^{\text{Node}(r)} \lambda_l \cdot \mathbb{I}(x \in L_l(r)), \tag{1}$$

where $\text{Node}(r)$ denotes the number of leaf nodes in the tree, $L_l(r)$ denotes the $l$-th leaf node containing a set of $x$ from $\mathcal{D}_{train}$, $\mathbb{I}(\cdot)$ is an indicator function determining if $x$ of interest would be assigned to leaf node $L_l(r)$ based on rule $r$, and $\lambda_l : \mathbb{X} \to \mathbb{Y}$ represents leaf-specific prediction values. Typically, $\lambda_l$ will be the average of corresponding labels for the leaf node in regression tasks, or the empirical class distribution in classification tasks. The structure of the tree is governed by a feature index vector $\ell$ and a threshold vector $\tau$, which together define the decision path containing a series of internal nodes from the root to each leaf node. Specifically, decision tree is constructed by recursively applying binary tests of the form $x^{\ell_j} \leq \tau_j$ at internal nodes, where $j$ indexes internal node along the root-to-leaf path, $\ell_j$ specifies feature index for $j$-th internal node split, $x^{\ell_j}$ denotes the corresponding feature value, and $\tau_j$ defines corresponding splitting threshold. In this work, we adopt CART [30] as our decision tree implementation due to two primary reasons: (i) it offers high interpretability and can be expressed using a simple if-else syntax, (ii) tree-based models, often ensembles of simple decision trees like CART, outperform deep learning approaches in numerous tabular prediction tasks [11, 5].

## 4 Proposed method: DeLTa

We propose a novel approach, Decision Tree Enhancer with LLM-derived Rule for Tabular Prediction (DeLTa), understanding the underlying rule logic encoded by decision trees and generating improved rules for tabular prediction via LLM, without requiring any fine-tuning of the LLM. This strategy introduces a fundamentally distinct interface for extending LLM capabilities to tabular tasks, departing from the majority of conventional data-to-text serialization pipelines that enforce LLMs to understand the serialized tabular data. In the following, we will elaborate on LLM-based decision tree rules refinement in Section 4.1; then, we will provide the calibration method for original decision trees via new generated rule by LLM in Section 4.2; we also give the overall implementation of ours in Section 4.3. An overview of the proposed framework is depicted in Fig. 1.

### 4.1 LLM-based decision tree rules refinement

The decision tree rules refinement procedure consists of the following stages: (i) decision tree rules initialization, that constructs a diverse decision tree rule set, and (ii) rule understanding via LLM, that leverages the LLM to understand and refine the original rule set to obtain a new rule.

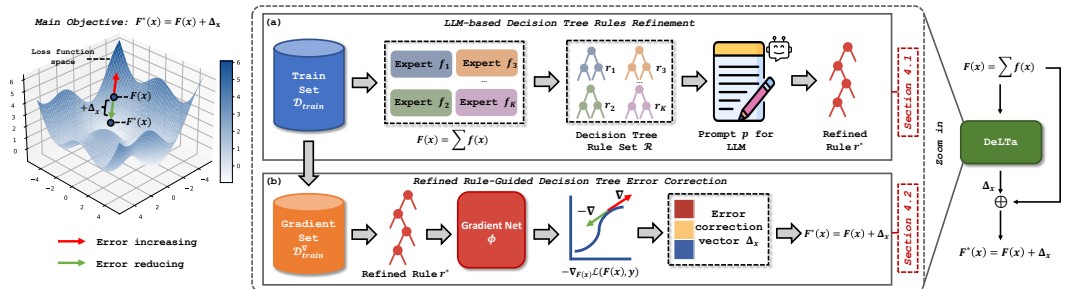

Figure 1: The DeLTa framework. As shown in the main objective, we calibrate the output of original decision tree experts $F(x)$ in the direction of "errors" reducing. Subfig (a) describes the process of refining decision tree rules with LLM, and subfig (b) details the refined rule-guided error correction for decision trees.

**Decision tree rules initialization.** To mitigate the risk of overfitting for a single decision tree expert [50], the classical Random Forest [51] algorithm trains diverse decision tree experts $\{f_k\}_{k=1}^K$ on different subsets of $\mathcal{D}_{train}$, and uses the ensemble of $\{f_k\}_{k=1}^K$ as prediction:

$$F(x) = \frac{1}{K} \sum_{k=1}^{K} f_k(x|\mathcal{D}_{train}^k, r_k), \tag{2}$$

where $F(x)$ denotes the label prediction produced by the Random Forest, $K$ is the number of experts, $\mathcal{D}_{train}^k$ denotes the subset, $r_k$ denotes the rule for expert $f_k$ derived from Eq. 1, and $\mathcal{R} = \{r_k\}_{k=1}^K$ denotes a source decision tree rule set. In terms of decision tree rule, each rule represents a logical comparison between features and thresholds displayed in semantically rich syntax, which thus can be used to partition the feature space into disjoint regions (i.e., leaf nodes) with high interpretability. A well understanding of these rules could generate a better feature space partitioning method, thereby promoting statistical coherence among samples assigned to the same leaf node. Although Random Forest ensembles the outputs from all rules, the inherent relationships and interactions among rules in $\mathcal{R}$ are ignored. With the development of $K$, analyzing these independent rules is gradually becoming more and more difficult, let alone utilizing these rules to partition feature space. To this end, we propose to leverage LLMs to analyze and summarize the rule set $\mathcal{R}$ into a refined rule due to the powerful logical reasoning ability of LLM.

**Rule understanding via LLM.** Our objective is to use an LLM to generate a new rule $r^*$ given original rule set $\mathcal{R}$. Specifically, we construct a prompt $p$ for the LLM that guides it to understand the diverse decision tree rule set $\mathcal{R}$ and synthesize the redesigned rule from the aggregated diverse knowledge. To effectively leverage the reasoning ability of LLM, the prompt $p$ (please see Appendix A.4) is designed to include: meta information $p_{meta}$ that describes the task objective, the extracted decision tree rule description $p_{rule}$ that contains $\mathcal{R}$, and the requirement $p_{requirement}$ for rule refinement. The formulation of generating $r^*$ is given by:

$$r^* = LLM(p) = LLM(p_{meta} \oplus p_{rule} \oplus p_{requirement}), \tag{3}$$

where $\oplus$ is the concatenation of each prompt, and this process is achieved through querying the LLM, without requiring any fine-tuning. Notably, our approach does not require additional domain-specific expertise or semantic information, such as explicit feature names or detailed task background knowledge. This makes it applicable to scenarios where such information is unavailable or incomplete.

Instead of relying on the intrinsic domain knowledge of tabular tasks, such as those in healthcare or finance, we frame the problem purely as a domain knowledge-agnostic machine learning task. Our method leverages LLMs to reason decision tree rule set $\mathcal{R}$ into an rule $r^*$, rather than directly inferring from serialized tabular data. To verify that LLM could generate a better rule, where samples grouped within the same leaf node exhibit greater statistical similarity, we compute the intra-node sample distance over all leaf nodes partitioned by $r^*$, and observe that the distance of $r^*$ is lower than original rule $r \in \mathcal{R}$, as illustrated in Fig. 2. Next, we will elaborate on how to leverage the new rule $r^*$ generated by LLM to enhance tabular predictions.

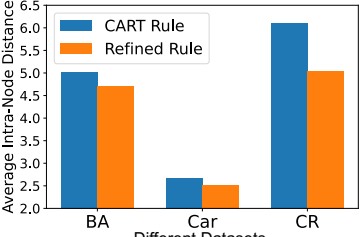

Figure 2: Average intra-node distance comparison.

## 4.2 Refined rule-guided decision tree error correction

The refined rule $r^*$ is summarized over the rule set $\mathcal{R}$, and thus provides a potentially better strategy that partitions the feature space of $\mathcal{D}_{train}$ into disjoint leaf nodes $\{L_l(r^*)\}_{l=1}^{\text{Node}(r^*)}$, with each leaf node containing a subset of $x$ from $\mathcal{D}_{train}$ and $\text{Node}(r^*)$ denoting the number of leaf nodes. To use $r^*$ for making predictions, the simplest strategies include: (1) treating it as a standalone decision tree by assigning the input $x$ to a specific leaf node, or (2) appending it to the existing ensemble of decision trees $F(x)$. However, the former may be too isolated, as it does not leverage the predictive power of the existing ensemble, while the latter may be too diluted, as it does not fully utilize the refined guidance provided by LLMs. This work introduces a novel way to leverage the power of $r^*$ together with $F(x)$ inspired by the intuition that estimating the difference between predictions and labels, i.e., the residual errors is often easier than directly predicting the ground-truth labels [52, 14]. Intuitively, we first use $r^*$ to partition data samples into its leaf nodes, expecting the samples in the same node are similar to each other. Motivated by the famous Gradient Boosting algorithm [53, 54], for the samples in each node, we learn a specific mapping function that predicts residual errors of $F(x)$ for the samples in that node. These learned functions are subsequently used to adjust the predictions made by $F(x)$, leading to improved accuracy. Since $r^*$ is designed to group similar samples effectively, the residuals within each leaf node are expected to be more structured and thus easier to predict.

Formally, we introduce the definition of the direction of prediction "errors" increasing and reducing, and the definition of a new set derived from $\mathcal{D}_{train}$ used for fitting the mapping functions as follows.

**Definition 1. Prediction "errors" increasing and reducing.** For a given training sample $x$, we approximate the "error" using the gradient $\nabla_{F(x)}\mathcal{L}(F(x), y)$, where $\mathcal{L}$ is the loss function discussed in Section 3. Intuitively, shifting $F(x)$ in the direction of this gradient (*"Errors" Increasing*) would likely worsen the model's performance, analogous to how traditional gradients indicate a direction in parameter space that increases the loss [55]. Therefore, it is natural that we should calibrate $F(x)$ in the direction of negative gradient $-\nabla_{F(x)}\mathcal{L}(F(x), y)$ (*"Errors" Reducing*).

**Definition 2. Gradient set $\mathcal{D}_{train}^{\nabla}$.** $\mathcal{D}_{train}^{\nabla} = \{(x, -\nabla_{F(x)}\mathcal{L}(F(x), y)) | (x, y) \in \mathcal{D}_{train}\}$ is derived from train set $\mathcal{D}_{train}$ and stores training sample features and their corresponding negative gradients of the loss function with respect to the output of $F(x)$. The only difference between $\mathcal{D}_{train}$ and $\mathcal{D}_{train}^{\nabla}$ lies in the labels, which are replaced by the negative gradients in $\mathcal{D}_{train}^{\nabla}$, while $r^*$ induces the same leaf node partitioning of the feature space for both datasets.

After obtaining the Gradient Set $\mathcal{D}_{train}^{\nabla}$, we apply a learnable Gradient Net $\phi(x|\mathcal{D}_{train}^{\nabla}, r^*)$ to model the mapping between $x$ and the corresponding negative gradient within $\mathcal{D}_{train}^{\nabla}$, where each leaf node $L_l(r^*)$ corresponds to one specific mapping function $\phi_l(\cdot; \theta_l) : x \to -\nabla_{F(x)}\mathcal{L}(F(x), y)$. The training process for $\phi_l$ is as follow:

$$\min_{\theta_l} \sum_{x \in L_l(r^*)} \|\phi_l(x; \theta_l) - (-\nabla_{F(x)}\mathcal{L}(F(x), y))\|_2^2, \tag{4}$$

where $\phi_l$ could be easily implemented by conventional machine learning model, such as CART [30], and $\phi_l$ over different leaf nodes are trained separately.

At inference time, given $x$ from $\mathcal{D}_{test}$, the prediction of Gradient Net $\phi$ could be produced as a sample-specific error correction vector $\Delta_x$. The final prediction $F^*(x)$ could be obtained by adding $\Delta_x$ to $F(x)$, that moves $F(x)$ in the direction of negative gradient ("Errors" Reducing) by one step:

$$\Delta_x = \eta * \phi(x|\mathcal{D}_{train}^{\nabla}, r^*) = \eta * \sum_{l=1}^{\text{Node}(r^*)} \phi_l(x; \theta_l) \cdot \mathbb{I}(x \in L_l(r^*)), \tag{5}$$

$$F^*(x) = F(x) + \Delta_x, \tag{6}$$

where $\mathbb{I}(\cdot)$ is an indicator function determining if $x$ of interest would be assigned to leaf node $L_l(r^*)$ based on rule $r^*$, $\eta \in \mathbb{R}_+$ is the hyperparameter similar to learning rate controlling the step size, and $\Delta_x$ approximates $-\eta * \nabla_{F(x)}\mathcal{L}(F(x), y)$. We now explain the proposed error correction strategy by giving a closer look at Eq. 6.

**Proposition 1** *Let $\mathbb{E}[\mathcal{L}(F(x), y)]$ denote the expected loss, where $F(x) = \frac{1}{K}\sum_k^K f_k(x|\mathcal{D}_{train}^k, r_k)$ and each $f_k$ corresponds to a decision tree rule $r_k$ from the expert-derived rule set $\mathcal{R} = \{r_k\}_{k=1}^K$.*

*Given a prompt $p$ that contains $\mathcal{R}$, the expected loss $\mathbb{E}\left[\mathcal{L}(F(x), y)\right]$ could be decreased by querying the LLM with $p$ to generate a refined rule $r^*$, which enables approximation of a sample-specific error correction vector $\Delta_x$ to guide prediction $F(x)$ in the direction of "errors" reducing.*

The proof of Proposition 1 is provided in Appendix A.5. To reduce the expected loss, we can approximate and apply negative gradient-based error correction vector $\Delta_x$ to $F(x)$.

### 4.3 Framework overview and discussion

**Framework overview.** In terms of training stage, the procedure begins by training a set of different decision tree experts (using CART for each expert) and obtaining the ensemble predictions based on Eq. 2. We then extract the corresponding decision tree rule set $\mathcal{R}$. The set of rules is used to construct a prompt $p$ for LLM, which returns an improved rule $r^*$ (Eq. 3). Subsequently, we compute the negative gradient for each training sample to construct the Gradient Set $\mathcal{D}_{train}^{\nabla}$, and then train the Gradient Net $\phi$ based on $\mathcal{D}_{train}^{\nabla}$ and $r^*$ (Eq. 4). During the test stage, we can achieve the preliminary prediction result $F(x)$ for each test sample $x$ based on the random forest; and also compute the sample-specific error correction vector $\Delta_x$ by feeding the test sample to Gradient Net. Now the final prediction can be corrected using Eq. 6. The complete training and inference pipeline is summarized in Algorithm 1 of Appendix A.3.

**Discussion.** To sum up, fed with the expert-derived rules $\mathcal{R}$, LLM is prompted to summarize multiple rules and redesign a new rule $r^*$, which thus considers the inherent relationships and interactions among multiple independent rules. Benefiting from the rule-based prompt, we neither need to fine-tune LLM nor expose the serialized samples into LLM. Due to the powerful reasoning ability of LLM, the refined rule $r^*$ that summarizes multiple rules is more effective and also convenient for us to use. For example, we can not only use the rule to make decision but also to partition the feature space of training set. The latter means that we can build a set of disjoint leaf nodes, where samples grouped within the same leaf node exhibit greater statistical similarity and thus can be used to train network. Here, we use the samples in the same leaf node to model the mapping between sample and the corresponding negative gradient, which is usually easier than directly fitting the relationship between samples and the labels. Our proposed DeLTa enables a principled integration of expert knowledge and LLM-based rule synthesis for enhanced performance, which paves a new way for LLM-based tabular data learning.

## 5 Experiments

**Datasets.** We consider a variety of supervised tabular prediction tasks with heterogeneous features, including binary classification, multiclass classification, and regression. Specifically, the tabular datasets include: Blood (BL) [56], Credit (CR) [57], Car [58], Bank (BA) [59], Adult (AD) [60], Jannis (JA) [11], Cpu_act (CP) [11], Credit_reg (CRR) [11], California_housing (CA) [61], House_16H (HO) [11], Fried (FR) [62], Diomand (DI) [63]. The dataset properties and data pre-processing details are summarized in Appendix A.1. Following previous studies [5], we use Accuracy (higher is better) to evaluate binary and multiclass classification tasks, Normalized Root Mean Squared Error (NRMSE) (lower is better) to evaluate the regression tasks.

**Baselines and implementation details.** We conduct a comparative analysis between DeLTa and other prominent methods in the field of tabular prediction. Specifically, we include LLM-based methods such as TabLLM [20], LIFT [19], TP-BERTa [21], GTL [22], P2T [23], FeatLLM [24]. For LIFT, we employ two versions, i.e., fine-tuning LLMs (LIFT) and in-context learning (LIFT-ICL) versions, as provided in their original paper. In addition, we compare DeLTa with non LLM-based methods, including KNN [64], CART [30], MLP [65], Random Forest [51], XGBoost [8], CatBoost [10], FT-Transformer [5], TabPFN [42], ModernNCA [38]. More details about the baseline models can be found in Appendix A.2. Implementation details of DeLTa including hyper-parameters are provided in Appendix A.3. For LLM usage, DeLTa adopts GPT-4o as its LLM backbone, yet is designed to be agnostic to the choice of LLMs (see Appendix A.6 for more results). To prevent LLMs from generating task-irrelevant output, we query LLM 10 times by default and use the average. In replicating baselines, GPT-4o is utilized for in-context learning. For baselines require LLM fine-tuning, we employ the LLMs used in their original paper: TabLLM uses T0 [46]; we use the API provided by OpenAI to perform black-box GPT-3.5 fine-tuning for LIFT, where the fine-tuning

Table 2: Test Accuracy (↑) performance of DeLTa and LLM-based baseline methods on classification tasks. "# Num" indicates the number of training samples. "Tab" indicates whether the LLM requires access to tabular samples. "Extra" indicates whether the method needs to use extra data samples outside the training set. "Feat" indicates whether the feature names are needed. "Tune" indicates whether LLM fine-tuning is needed.

| # Num | Method (LLM used) | Additional requirements | | | | Datasets | | | | | | |
| | | Tab | Extra | Feat | Tune | BL ↑ | CR ↑ | Car ↑ | BA ↑ | AD ↑ | JA ↑ | Average ↑ |
|---|---|---|---|---|---|---|---|---|---|---|---|---|
| All | TabLLM (T0) | ✓ | – | ✓ | ✓ | 0.761 | 0.737 | **0.845** | 0.906 | 0.864 | 0.691 | 0.801 |
| All | LIFT (GPT-3.5) | ✓ | – | ✓ | ✓ | 0.689 | 0.691 | 0.729 | 0.825 | 0.81 | 0.647 | 0.732 |
| All | TP-BERTa (RoBERTa) | ✓ | ✓ | ✓ | ✓ | 0.761 | 0.730 | 0.826 | **0.916** | 0.844 | 0.659 | 0.789 |
| All | GTL (LLaMA2) | ✓ | ✓ | ✓ | ✓ | N/A | N/A | N/A | N/A | N/A | N/A | N/A |
| All | LIFT-ICL (GPT-4o) | ✓ | – | ✓ | – | N/A | N/A | N/A | N/A | N/A | N/A | N/A |
| All | P2T (GPT-4o) | ✓ | ✓ | ✓ | – | N/A | N/A | N/A | N/A | N/A | N/A | N/A |
| All | FeatLLM (GPT-4o) | ✓ | – | ✓ | – | 0.768 | 0.701 | 0.589 | 0.887 | 0.842 | 0.540 | 0.721 |
| All | **DeLTa** (GPT-4o) | – | – | – | – | **0.829** | **0.783** | 0.836 | 0.908 | **0.868** | **0.705** | **0.822** |
| 256 | TabLLM (T0) | ✓ | – | ✓ | ✓ | 0.344 | 0.690 | 0.471 | 0.785 | **0.787** | 0.373 | 0.575 |
| 256 | LIFT (GPT-3.5) | ✓ | – | ✓ | ✓ | 0.651 | 0.682 | 0.431 | 0.724 | 0.773 | 0.398 | 0.610 |
| 256 | TP-BERTa (RoBERTa) | ✓ | ✓ | ✓ | ✓ | 0.721 | 0.677 | 0.408 | **0.853** | 0.772 | 0.464 | 0.649 |
| 256 | GTL (LLaMA2) | ✓ | ✓ | ✓ | ✓ | N/A | N/A | N/A | N/A | N/A | N/A | N/A |
| 256 | LIFT-ICL (GPT-4o) | ✓ | – | ✓ | – | N/A | N/A | N/A | N/A | N/A | N/A | N/A |
| 256 | P2T (GPT-4o) | ✓ | ✓ | ✓ | – | N/A | N/A | N/A | N/A | N/A | N/A | N/A |
| 256 | FeatLLM (GPT-4o) | ✓ | – | ✓ | – | 0.729 | 0.658 | 0.459 | 0.740 | 0.772 | 0.473 | 0.639 |
| 256 | **DeLTa** (GPT-4o) | – | – | – | – | **0.732** | **0.713** | **0.736** | 0.767 | 0.778 | **0.476** | **0.700** |
| 64 | TabLLM (T0) | ✓ | – | ✓ | ✓ | 0.245 | **0.673** | 0.386 | 0.602 | **0.785** | 0.268 | 0.493 |
| 64 | LIFT (GPT-3.5) | ✓ | – | ✓ | ✓ | 0.643 | 0.643 | 0.500 | 0.728 | 0.748 | 0.430 | 0.615 |
| 64 | TP-BERTa (RoBERTa) | ✓ | ✓ | ✓ | ✓ | 0.648 | 0.646 | 0.419 | **0.840** | 0.761 | 0.414 | 0.621 |
| 64 | GTL (LLaMA2) | ✓ | ✓ | ✓ | ✓ | 0.613 | 0.638 | 0.512 | 0.719 | 0.750 | 0.414 | 0.608 |
| 64 | LIFT-ICL (GPT-4o) | ✓ | – | ✓ | – | 0.348 | 0.538 | 0.326 | 0.469 | 0.613 | 0.251 | 0.424 |
| 64 | P2T (GPT-4o) | ✓ | ✓ | ✓ | – | 0.368 | 0.542 | 0.334 | 0.569 | 0.634 | 0.243 | 0.448 |
| 64 | FeatLLM (GPT-4o) | ✓ | – | ✓ | – | 0.697 | 0.643 | 0.508 | 0.723 | 0.749 | 0.450 | 0.628 |
| 64 | **DeLTa** (GPT-4o) | – | – | – | – | **0.732** | 0.663 | **0.615** | 0.733 | 0.758 | **0.457** | **0.660** |

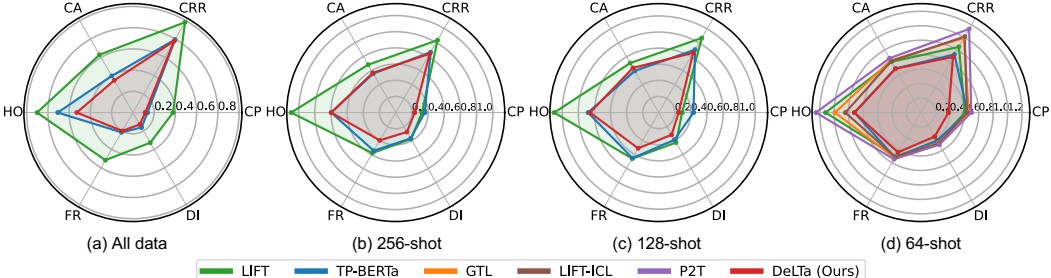

(a) All data  (b) 256-shot  (c) 128-shot  (d) 64-shot

LIFT — TP-BERTa — GTL — LIFT-ICL — P2T — DeLTa (Ours)

Figure 3: Test NRMSE (↓) performance of DeLTa and LLM-based baseline methods on regression tasks.

method is suggested by the original paper; TP-BERTa employs RoBERTa [66]; GTL uses the 13B version of LLaMA 2 [67]. The experiments are run on NVIDIA A100-PCIE-40GB GPU.

## 5.1 Main Results

**Comparison with LLM-based methods.** We evaluated the performance of DeLTa against mainstream LLM-based baseline methods on tabular classification tasks, as illustrated in Table 2. In addition to standard prediction tasks under full-data regimes, we also provide results under low-data regimes (few-shot learning), as some of these LLM-based baseline methods are constrained by the limited input length of LLMs and are thus restricted to low-data settings. Following prior work [24], we simulate the few-shot setting by randomly sampling a fixed number of training examples, where the number of shots denotes the total number of selected samples. Considering some of the LLM-based baseline methods are applicable to regression tasks, we further evaluate DeLTa on regression under both full-data and low-data regimes settings, as illustrated in Fig. 3. Overall, DeLTa achieves the highest average performance in all settings, including classification and regression, in both full and low-data regimes. While DeLTa is primarily designed for full-data tabular prediction, our results demonstrate that it also performs competitively in low-data scenarios. Furthermore, compared to existing LLM-based tabular prediction methods, the proposed DeLTa requires no additional requirements for LLMs as illustrated in Table 2. It also aligns with our claims that we have solved the two inherent issues for prior LLM-based approaches: (i) the proposed DeLTa avoids reliance on column names and tabular data serialization, and thus will protect sample-level privacy information, and (ii) rather than directly using LLMs to generate label predictions, DeLTa utilizes the powerful reasoning

Table 3: Runtime in seconds of DeLTa and other LLM-based baseline methods for the training and inference phase, conducted on Adult dataset. "# Num" indicates number of training samples. †These methods require querying LLMs for each test sample. * These methods do not require querying LLMs at inference phase.

| # Num | Stage | TabLLM† | LIFT† | TP-BERTa† | GTL† | LIFT-ICL† | P2T† | FeatLLM* | **DeLTa*** (Ours) |
|---|---|---|---|---|---|---|---|---|---|
| All | Train | 177371.37 | 153689.14 | 1319.35 | N/A | N/A | N/A | 1231.22 | **35.20** |
| | Inference | 179.21 | 90149.48 | 1.58 | N/A | N/A | N/A | 0.14 | **0.09** |
| 64 | Train | 288.61 | 191.75 | 316.40 | N/A | N/A | N/A | 1047.55 | **23.09** |
| | Inference | 170.42 | 89684.76 | 5.01 | 153501.02 | 176726.13 | 200258.24 | 0.14 | **0.08** |

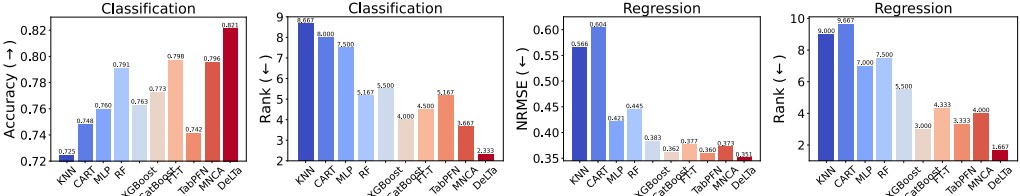

Figure 4: Test performance of DeLTa and non-LLM baseline methods on classification and regression tasks. Here, "RF" denotes Random Forest, "FT-T" denotes FT-Transformer, "MNCA" denotes ModernNCA.

ability of LLMs to enhance decision trees, enabling effective tabular prediction in both full-data and low-data regimes without requiring LLM fine-tuning.

**Computational efficiency.** We compare the computational cost between ours and LLM-based methods in Table 3. The results show that our proposed DeLTa is computationally efficient in both training and inference stages. This efficiency stems from key properties: (i) DeLTa utilizes the reason ability of LLM without fine-tuning LLM, (ii) DeLTa avoids querying LLMs to generate predictions for individual samples. Instead, it only needs to query LLM via API to generate one refined decision tree rule for one dataset, enabling significantly more efficient use of LLM resources.

**Comparison with conventional methods.** We further compare the performance of DeLTa and other conventional tabular models on full data regimes averaged over all datasets, as illustrated in Fig. 4. Full results are provided in Appendix A.8. The results show that DeLTa also performs well compared with conventional tabular models on both classification and regression tasks.

## 5.2 Further analysis

**Ablation study.** We further conduct ablation study to demonstrate the effectiveness of key components of DeLTa, as illustrated in Table 4, by comparing DeLTa against three variants: (i) DeLTa w/o "RR", that replaces the LLM generated $r^*$ used in Gradient Net $\phi(x|\mathcal{D}_{train}^{\nabla}, r^*)$ with $r \in \mathcal{R}$ derived from Random Forest $F(x)$, (ii) DeLTa w/o "EC", that directly obtains the label prediction for test $x$ by utilizing $f(x|\mathcal{D}_{train}, r^*)$ (Eq. 1) with the refined rule $r^*$, (iii) Random Forest, which makes decision with $F(x)$ in Eq. 2. The results show that the decision tree rule refinement via LLM is important, and the error correction strategy could further enhance tabular prediction performance.

Table 4: Ablation study on the different components in DeLTa; see full results in Appendix A.10. "RR" denotes the rule refinement process, and "EC" denotes the error correction process.

| Variants | CR ↑ | CA ↓ |
|---|---|---|
| **DeLTa** (Ours) | **0.7829** | **0.3507** |
| w/o "RR" | 0.7257 | 0.5443 |
| w/o "EC" | 0.7657 | 0.4016 |
| Random Forest | 0.7371 | 0.5057 |

**Additional results.** Fig. 5 shows the label predictions of ours: $F(x) + \Delta_x$ and ours w/o $\Delta_x$: $F(x)$. We can observe that two categories of samples are overlapped in some regions. $F(x)$ struggles to distinguish them, but DeLTa could correct the prediction errors of $F(x)$ to enhance the prediction for such complex data patterns.

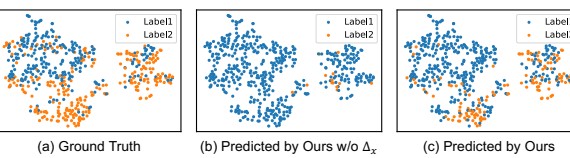

Figure 5: Visualization of label prediction of DeLTa w/ and w/o error correction vector $\Delta_x$ on BA dataset.

We incorporate the sensitivity analysis for hyperparameters in Appendix A.11. Additional insights and analysis regarding the LLM-refined rule can be found in Appendix A.12.

## 6   Conclusion

In this paper, we propose DeLTa, a decision tree enhancer with LLM-derived rule for tabular prediction, to solve the two challenges w.r.t. data perspective and model perspective in prior LLM-based approaches for tabular prediction. DeLTa avoids reliance on feature names and tabular data serialization, and it utilizes the powerful reasoning ability of LLMs to enhance decision trees, enabling effective tabular prediction in both full-data and low-data regimes without requiring LLM fine-tuning. The empirical results demonstrated the effectiveness of DeLTa for tabular prediction tasks. Our work can shed some light on developing better algorithms for similar tasks.

## Acknowledgments and Disclosure of Funding

The authors would like to thank the anonymous referees for their valuable comments. In this work, Hangting Ye, Jinmeng Li, Dandan Guo and Yi Chang are supported by the National Key R&D Program of China under Grant (No. 2023YFF0905400) and the National Natural Science Foundation of China (No. 623B2043, No. U2341229, No. 62306125).

Part of this work was carried out during Dandan Guo's visit to King Abdullah University of Science and Technology (KAUST).

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

# A  Technical Appendices and Supplementary Material

Technical appendices with additional results, figures, graphs and proofs may be submitted with the paper submission before the full submission deadline (see above), or as a separate PDF in the ZIP file below before the supplementary material deadline. There is no page limit for the technical appendices.

## A.1  Datasets details

We consider a variety of supervised tabular prediction tasks with heterogeneous features, including classification and regression. Specifically, the tabular datasets include: Blood (BL), Credit (CR), Car, Bank (BA), Adult (AD), Jannis (JA), Cpu_act (CP), Credit_reg (CRR), California_housing (CA), House_16H (HO), Fried (FR), Diomand (DI). The dataset properties are summarized in Table 5. Our primary objective was to evaluate DeLTa's effectiveness in the context of LLM-based tabular prediction, and to ensure fair comparison with existing methods in this category. Accordingly, the majority of our classification datasets were selected based on their widespread use in recent LLM-based tabular learning studies such as TabLLM and FeatLLM, both of which primarily focus on classification tasks. To extend our evaluation beyond classification, we also incorporated regression datasets from the TALENT benchmark [68, 69], which has recently gained attention for assessing tabular regression performance. To ensure fair comparisons, we adhere to identical preprocessing procedures following TALENT benchmark [69] for each dataset.

Table 5: Tabular dataset properties. "#objects" indicates the number of samples in the dataset. "#num. features" indicates the number of numerical features, and "#cat. features" indicates the number of categorical features.

|  | BL | CR | Car | BA | AD | JA | CP | CRR | CA | HO | FR | DI |
|---|---|---|---|---|---|---|---|---|---|---|---|---|
| #objects | 748 | 1000 | 1728 | 45211 | 48842 | 83733 | 8192 | 16714 | 20640 | 22784 | 40768 | 53940 |
| #num. features | 4 | 7 | 0 | 7 | 6 | 54 | 21 | 10 | 8 | 16 | 10 | 6 |
| #cat. features | 0 | 13 | 6 | 9 | 8 | 0 | 0 | 0 | 0 | 0 | 0 | 3 |
| metric | Acc. | Acc. | Acc. | Acc. | Acc. | Acc. | NRMSE | NRMSE | NRMSE | NRMSE | NRMSE | NRMSE |
| #classes | 2 | 2 | 4 | 2 | 2 | 4 | – | – | – | – | – | – |

## A.2  Baseline models details

We include LLM-based methods such as TabLLM, LIFT, LIFT-ICL, TP-BERTa, GTL, P2T, FeatLLM: (1) TabLLM employs T0 as the base LLM and investigates the performance of fine-tuned T0 model in very few-shot scenarios; (2) LIFT investigates the performance of fine-tuned GPT models on tabular data; (3) LIFT-ICL is another version of LIFT provided in the original paper, that utilizes in-context learning without fine-tuning by providing a few training examples; (4) TP-BERTa introduces a tabular-specific tokenization scheme, pretraining a single language model across multiple tabular datasets. Given target tabular datasets, the pretrained language model should be further fine-tuned; (5) GTL pretrains LLaMA across multiple tabular datasets to achieve zero-shot and in-context learning on target tabular dataset; (6) P2T leverages extra unlabeled samples outside the training set as demonstrations to conduct in-context learning; (7) FeatLLM prompts LLMs with a few training examples to generate new features to replace original features and then trains linear models for few-shot classification tasks. In replicating baselines, GPT-4o is utilized for in-context learning. For baselines require LLM fine-tuning, we employ the LLMs used in their original paper: TabLLM uses T0; we use the API provided by OpenAI to perform black-box GPT-3.5 fine-tuning for LIFT, where the fine-tuning method is suggested by the original paper; TP-BERTa employs RoBERTa; GTL uses the 13B version of LLaMA 2. The implementation of these methods is based on their official open-source code releases.

In addition, we compare DeLTa with non LLM-based methods, including KNN, CART, MLP, Random Forest, XGBoost, CatBoost, ResNet, FT-Transformer, Saint, TabPFN and ModernNCA. Among them, (1) KNN classifies or predicts a sample based on the majority vote or average of its $k$ nearest neighbors in the feature space; (2) CART is an implementation for the simplest decision trees; (3) MLP is a type of feedforward neural network composed of multiple layers; (4) Random Forest, (5) XGBoost and (6) CatBoost are the variants of decision tree-based methods; (7) ResNet is a deep

neural network architecture that introduces skip connections to enable the training of very deep networks; (8) FT-Transformer is a token-based method which transforms features to embeddings and applies a series of attention-based transformations to the embeddings; (9) Saint is a token-based method that leverages row and column attention mechanisms for tabular data; (10) TabPFN generates a synthetic dataset with diverse distribution to pretrain a model, and then leverages the training samples of target dataset for inference. We use the version 2 of TabPFN since it can be applied to both classification and regression datasets; (11) ModernNCA makes predictions based on the relationships with neighbors in a learned embedding space. The implementation of these methods is based on the official open-source code releases from TALENT benchmark [69].

## A.3 DeLTa details

---

**Algorithm 1** DeLTa training and inference workflow.

---

**Input:** Training set $\mathcal{D}_{train}$, test set $\mathcal{D}_{test}$ (without accessing to label), number of decision tree experts $K$, API of LLM;
Initialize source decision tree rule set $\mathcal{R} = []$;
**for** $k = 1$ **to** $K$ **do** // This loop can be executed in parallel.
    Train one decision tree expert $f_k$ on $\mathcal{D}_{train}^k$ with Eq. 1;
    Extract the corresponding rule $r_k$ of $f_k$;
    Update $\mathcal{R} \leftarrow \mathcal{R} \cup r_k$
**end for**
Obtain the ensemble prediction of decision tree experts $F(x)$ with Eq. 2;
Form prompt $p$ based on $\mathcal{R}$;
Query LLM with prompt $p$ and generate the new rule $r^*$ with Eq. 3;
Calculate Gradient Set $\mathcal{D}_{train}^{\nabla} = \{(x, -\nabla_{F(x)}\mathcal{L}(F(x), y)) | (x, y) \in \mathcal{D}_{train}\}$;
Train Gradient Net $\phi$ based on Gradient Set with Eq. 4;
**for** $x$ in $\mathcal{D}_{test}$ **do** // Without accessing to label. And this loop can be executed in parallel.
    Estimate error correction vector $\Delta_x = \eta * \phi(x|\mathcal{D}_{train}^{\nabla}, r^*)$ with Eq. 5;
    Computing final prediction $F^*(x) = F(x) + \Delta_x$ with Eq. 6;
**end for**
**Return:** Final prediction $\{F^*(x) | x \in \mathcal{D}_{test}\}$.

---

The DeLTa architecture remains consistent across all datasets. Specifically, the $F(x)$ is produced by the output of Random Forest. For LLM usage, DeLTa adopts GPT-4o via API as its LLM backbone, yet is designed to be agnostic to the choice of LLMs (see Table 6 and Table 7 for more results). To prevent LLMs from generating task-irrelevant output, we query LLM 10 times by default and use the average. DeLTa needs to construct Gradient Set $\mathcal{D}_{train}^{\nabla} = \{(x, -\nabla_{F(x)}\mathcal{L}(F(x), y)) | (x, y) \in \mathcal{D}_{train}\}$, which stores training sample features and their corresponding negative gradients of the loss function with respect to the output of $F(x)$. Here, $\mathcal{L}$ is a smooth loss function (e.g., mean squared error for regression or cross-entropy for classification). And $-\nabla_{F(x)}\mathcal{L}(F(x), y) \in \mathbb{R}$ for regression, $-\nabla_{F(x)}\mathcal{L}(F(x), y) \in \mathbb{R}$ for binary classification, and $-\nabla_{F(x)}\mathcal{L}(F(x), y) \in \mathbb{R}^c$ for multiclass classification, where $c$ is the number of classes. The Gradient Net $\phi$ contains learnable leaf node-specific mapping function $\phi_l$, where $\phi_l$ is implemented by CART for classification and TabPFN for regression, and $\phi_l$ over different leaf nodes are trained separately. The number of leaf node for LLM generated rule $r^*$ will less than 30, and different $\phi_l$ can be trained in parallel, which will result in a low runtime cost as illustrated in Table 14. The training and inference pipeline is summarized in Algorithm 1.

## A.4 Example prompt for refining decision tree rules

We leverage LLM to refine the decision tree rule set $\mathcal{R}$ to generate a better rule $r^*$, where samples grouped within the same leaf node exhibit greater statistical similarity. Therefore, we query LLM to generate a better rule to partition the feature space into disjoint regions (i.e., leaf nodes). Specifically, the prompt $p$ is designed to include: meta information $p_{meta}$ that describes the task objective, the extracted decision tree rule description $p_{rule}$ that contains $\mathcal{R}$, and the requirement $p_{requirement}$ for rule refinement. Please note that our approach does not require additional domain-specific expertise or semantic information, such as explicit feature names or detailed task background knowledge. This

makes it applicable to scenarios where such information is unavailable or incomplete. Instead of relying on the intrinsic domain knowledge of tabular tasks, such as those in healthcare or finance, we frame the problem purely as an domain knowledge-agnostic machine learning task. Our method leverages LLMs to reason decision tree rule set $\mathcal{R}$ into an rule $r^*$, rather than directly inferring from serialized tabular data, and thus will protect sample-level privacy information. The prompt example is provided in Fig. 6.

---

You are an expert in tabular machine learning domain. I will provide the meta information, the CART tree rules about the prediction task. Please help me design a better rule for inference.
# Meta information about dataset.

```
{
    "name": "adult",
    "task_type": "binclass",
    "n_num_features": 6,
    "n_cat_features": 8,
    "train_size": 26048,
}
```

# CART tree rules (We provide the example of decision rule of one subtree from the Random Forest below.)

```
----------------------------
Tree 1 rules:
|--- feature_10 <= -0.59
|   |--- feature_4 <= 4.31
|   |   |--- feature_2 <= 0.94
|   |   |   |--- feature_6 <= -1.82
|   |   |   |   |--- class: 1.0
|   |   |   |--- feature_6 >  -1.82
|   |   |   |   |--- class: 0.0
|   |   |--- feature_2 >  0.94
|   |   |   |--- feature_3 <= 0.53
|   |   |   |   |--- class: 1.0
|   |   |   |--- feature_3 >  0.53
|   |   |   |   |--- class: 1.0
|   |--- feature_4 >  4.31
|   |   |--- class: 1.0
|--- feature_10 >  -0.59
|   |--- feature_0 <= -0.74
|   |   |--- class: 0.0
|   |--- feature_0 >  -0.74
|   |   |--- feature_3 <= 0.80
|   |   |   |--- feature_2 <= 0.94
|   |   |   |   |--- class: 0.0
|   |   |   |--- feature_2 >  0.94
|   |   |   |   |--- feature_10 <= 1.89
|   |   |   |   |   |--- class: 0.0
|   |   |   |   |--- feature_10 >  1.89
|   |   |   |   |   |--- class: 1.0
|   |   |--- feature_3 >  0.80
|   |   |   |--- class: 1.0
----------------------------
Tree 2 rules:
...
----------------------------
```

# CART tree rules end

Based on the above information, please learn the rules evolving process and help me design a better rule like what cart used for inference to achieve higher performance. Please not just copy, please refine these rules and create a new better one. The new rule aims to divide the training space into several regions, where each region is denoted by a unique leaf node id. The number of leaf nodes should no more than $x$. Please return the dict format of rule, the format should be strictly like:

```
self.tree = {
        "feature": 11,
        "threshold": -0.78,
        "operator": "<=",
        "left": {"id": "leaf_1"},
        "right": {
            "feature": 7,
            "threshold": -0.46,
            "operator": "<=",
            "left": {"id": "leaf_2"},
            "right": {"id": "leaf_3"},
        },
        }
```

Please note that each leaf id node can only appear once, for example, `"id": "leaf_1"` can only appear once. Thus you only need to return the leaf nodes, rather than the true predictions.

---

Figure 6: An example of the prompt template for the adult dataset. Here, $x$ denotes the maximum number of leaf nodes among all the provided decision trees.

## A.5 Theoretical analysis

**Proposition 2** *Let $\mathbb{E}\left[\mathcal{L}(F(x), y)\right]$ denote the expected loss, where $F(x) = \frac{1}{K}\sum_k^K f_k(x|\mathcal{D}_{train}^k, r_k)$ and each $f_k$ corresponds to a decision tree rule $r_k$ from the expert-derived rule set $\mathcal{R} = \{r_k\}_{k=1}^K$. Given a prompt $p$ that contains $\mathcal{R}$, the expected loss $\mathbb{E}\left[\mathcal{L}(F(x), y)\right]$ could be decreased by querying the LLM with $p$ to generate a refined rule $r^*$, which enables approximation of a sample-specific error correction vector $\Delta_x$ to guide prediction $F(x)$ in the direction of "errors" reducing.*

*Proof.* The the expected loss $\mathbb{E}\left[\mathcal{L}(F(x), y)\right]$ after calibration is formulated as:

$$
\begin{aligned}
\mathbb{E}\left[\mathcal{L}(F^*(x), y)\right] &= \mathbb{E}\left[\mathcal{L}(F(x) + \Delta_x, y)\right] \\
&\approx \mathbb{E}\left[\mathcal{L}(F(x), y) + \Delta_x * \nabla_{F(x)}\mathcal{L}(F(x), y)\right] \\
&= \mathbb{E}\left[\mathcal{L}(F(x), y) + (-\eta * \nabla_{F(x)}\mathcal{L}(F(x), y)) * \nabla_{F(x)}\mathcal{L}(F(x), y)\right] \quad (7) \\
&= \mathbb{E}\left[\mathcal{L}(F(x), y) + (-\eta * \|\nabla_{F(x)}\mathcal{L}(F(x), y)\|_2^2)\right] \\
&< \mathbb{E}\left[\mathcal{L}(F(x), y)\right],
\end{aligned}
$$

where $\mathbb{E}\left[\mathcal{L}(F(x) + \Delta_x, y)\right] \approx \mathbb{E}\left[\mathcal{L}(F(x), y) + \Delta_x * \nabla_{F(x)}\mathcal{L}(F(x), y)\right]$ is given by Taylor first-order expansion and $\mathcal{L}$ is a convex loss function (e.g., mean squared error or cross-entropy). Error correction vector $\Delta_x = \eta * \phi(x|\mathcal{D}_{train}^\nabla, r^*)$ approximates the $-\eta * \nabla_{F(x)}\mathcal{L}(F(x), y)$ in the direction of "errors" reducing according to Eq. 4 in the original paper and $\eta \in \mathbb{R}_+$ is the hyperparameter. To reduce the expected loss, we can approximate and apply negative gradient-based error correction vector $\Delta_x$ to $F(x)$.

## A.6 Varying LLM backbones

To investigate the impact of using different LLMs in our framework, given their distinct prior knowledge and reasoning abilities from being trained on various text corpora, we measured performance using not only GPT-4o but also the Qwen3-32B (open-source) as a backbone for comparison. The results in Table 6 and Table 7 indicate that while certain tasks show greater improvement with specific LLMs, the average performance is similar with different LLMs.

Table 6: Effects of various LLM backbones: GPT-4o vs. Qwen3-32B on classification tasks ↑ in full-data and low-data regimes. "# Num " indicates the number of training samples.

| # Num | Various LLM backbones | BL ↑ | CR↑ | Car ↑ | BA ↑ | AD ↑ | JA ↑ | Average ↑ |
|---|---|---|---|---|---|---|---|---|
| All | DeLTa (Qwen3-32B) | 0.821 | 0.766 | 0.832 | 0.907 | 0.867 | 0.703 | 0.816 |
| All | DeLTa (GPT-4o) (Ours) | 0.829 | 0.783 | 0.836 | 0.908 | 0.868 | 0.705 | 0.822 |
| 256 | DeLTa (Qwen3-32B) | 0.741 | 0.707 | 0.750 | 0.765 | 0.778 | 0.474 | 0.703 |
| 256 | DeLTa (GPT-4o) (Ours) | 0.732 | 0.713 | 0.736 | 0.767 | 0.778 | 0.476 | 0.700 |
| 128 | DeLTa (Qwen3-32B) | 0.744 | 0.687 | 0.679 | 0.749 | 0.778 | 0.456 | 0.682 |
| 128 | DeLTa (GPT-4o) (Ours) | 0.746 | 0.683 | 0.722 | 0.749 | 0.774 | 0.457 | 0.688 |
| 64 | DeLTa (Qwen3-32B) | 0.736 | 0.657 | 0.567 | 0.729 | 0.759 | 0.454 | 0.650 |
| 64 | DeLTa (GPT-4o) (Ours) | 0.732 | 0.663 | 0.615 | 0.733 | 0.758 | 0.457 | 0.660 |

Table 7: Effects of various LLM backbones: GPT-4o vs. Qwen3-32B on regression tasks ↓ in full-data and low-data regimes. "# Num " indicates the number of training samples.

| # Num | Various LLM backbones | CP↓ | CRR ↓ | CA ↓ | HO↓ | FR↓ | DI↓ | Average ↓ |
|---|---|---|---|---|---|---|---|---|
| All | DeLTa (Qwen3-32B) | 0.114 | 0.788 | 0.367 | 0.541 | 0.201 | 0.129 | 0.357 |
| All | DeLTa (GPT-4o) (Ours) | 0.116 | 0.780 | 0.351 | 0.529 | 0.200 | 0.130 | 0.351 |
| 256 | DeLTa (Qwen3-32B) | 0.203 | 0.854 | 0.606 | 0.811 | 0.358 | 0.293 | 0.521 |
| 256 | DeLTa (GPT-4o) (Ours) | 0.234 | 0.841 | 0.568 | 0.799 | 0.397 | 0.277 | 0.519 |
| 128 | DeLTa (Qwen3-32B) | 0.206 | 0.876 | 0.709 | 0.940 | 0.474 | 0.348 | 0.592 |
| 128 | DeLTa (GPT-4o) (Ours) | 0.247 | 0.857 | 0.641 | 0.855 | 0.512 | 0.319 | 0.572 |
| 64 | DeLTa (Qwen3-32B) | 0.289 | 0.920 | 0.737 | 0.944 | 0.598 | 0.406 | 0.649 |
| 64 | DeLTa (GPT-4o) (Ours) | 0.382 | 0.888 | 0.701 | 0.919 | 0.634 | 0.383 | 0.651 |

## A.7 Full comparison results with LLM-based baseline methods

We provide the full results of DeLTa against mainstream LLM-based baseline methods in full-data and low-data regimes in Table 8 to Table 11. "# Num" indicates the number of training samples. "Tab" indicates whether the LLM requires access to tabular samples. "Extra" indicates whether the method needs to use extra data samples outside the training set. "Feat" indicates whether the feature names are needed. "Tune" indicates whether LLM fine-tuning is needed.

Table 8: Comparison with LLM-based methods averaged over all classification datasets ↑. Average standard deviations over all datasets are given.

| Methods | Additional requirements | | | | Classification, # Num | | | | | |
|---|---|---|---|---|---|---|---|---|---|---|
| | Tab | Extra | Feat | Tune | All | 512 | 256 | 128 | 64 | 16 |
| TabLLM (T0) | ✓ | – | ✓ | ✓ | $\underline{0.801}_{012}$ | $0.568_{035}$ | $0.575_{078}$ | $0.519_{046}$ | $0.493_{054}$ | $0.413_{082}$ |
| LIFT (GPT-3.5) | ✓ | – | ✓ | ✓ | $0.732_{051}$ | $0.625_{045}$ | $0.610_{056}$ | $\underline{0.650}_{064}$ | $0.615_{059}$ | $0.556_{083}$ |
| TP-BERTa (RoBERTa) | ✓ | ✓ | ✓ | ✓ | $0.789_{016}$ | $0.638_{029}$ | $\underline{0.649}_{063}$ | $0.641_{076}$ | $0.621_{073}$ | $0.563_{091}$ |
| GTL (LLaMA2) | ✓ | ✓ | ✓ | ✓ | N/A | N/A | N/A | N/A | $0.608_{058}$ | $0.552_{079}$ |
| LIFT-ICL (GPT-4o) | ✓ | – | ✓ | – | N/A | N/A | N/A | N/A | $0.424_{107}$ | $0.382_{081}$ |
| P2T (GPT-4o) | ✓ | ✓ | ✓ | – | N/A | N/A | N/A | N/A | $0.448_{060}$ | $0.304_{072}$ |
| FeatLLM (GPT-4o) | ✓ | – | ✓ | – | $0.721_{024}$ | $\underline{0.664}_{024}$ | $0.639_{041}$ | $0.646_{041}$ | $\underline{0.628}_{047}$ | $\underline{0.570}_{066}$ |
| **Ours** (GPT-4o) | – | – | – | – | $\mathbf{0.822}_{005}$ | $\mathbf{0.719}_{026}$ | $\mathbf{0.700}_{018}$ | $\mathbf{0.688}_{022}$ | $\mathbf{0.660}_{045}$ | $\mathbf{0.579}_{055}$ |

Table 9: Comparison with LLM-based methods averaged over all regression datasets ↓. Average standard deviations over all datasets are given.

| Methods | Additional requirements | | | | Regression, # Num | | | | | |
|---|---|---|---|---|---|---|---|---|---|---|
| | Tab | Extra | Feat | Tune | All | 512 | 256 | 128 | 64 | 16 |
| TabLLM (T0) | ✓ | – | ✓ | ✓ | N/A | N/A | N/A | N/A | N/A | N/A |
| LIFT (GPT-3.5) | ✓ | – | ✓ | ✓ | $0.623_{048}$ | $0.782_{056}$ | $0.715_{054}$ | $0.743_{047}$ | $0.827_{098}$ | $0.995_{095}$ |
| TP-BERTa (RoBERTa) | ✓ | ✓ | ✓ | ✓ | $\underline{0.403}_{015}$ | $\underline{0.496}_{011}$ | $\underline{0.582}_{019}$ | $\underline{0.642}_{036}$ | $\underline{0.727}_{028}$ | $\underline{0.887}_{028}$ |
| GTL (LLaMA2) | ✓ | ✓ | ✓ | ✓ | N/A | N/A | N/A | N/A | $0.847_{072}$ | $0.938_{087}$ |
| LIFT-ICL (GPT-4o) | ✓ | – | ✓ | – | N/A | N/A | N/A | N/A | $0.820_{084}$ | $1.025_{084}$ |
| P2T (GPT-4o) | ✓ | ✓ | ✓ | – | N/A | N/A | N/A | N/A | $0.935_{062}$ | $1.042_{064}$ |
| FeatLLM (GPT-4o) | ✓ | – | ✓ | – | N/A | N/A | N/A | N/A | N/A | N/A |
| **Ours** (GPT-4o) | – | – | – | – | $\mathbf{0.351}_{008}$ | $\mathbf{0.462}_{015}$ | $\mathbf{0.519}_{028}$ | $\mathbf{0.572}_{032}$ | $\mathbf{0.651}_{043}$ | $\mathbf{0.852}_{056}$ |

Table 10: Comparison with LLM-based methods on all classification datasets ↑. We also calculate the average relative improvement of DeLTa against the best baseline method (↑ %). "# Num" indicates the number of training samples. This table serves as an extension of Table 8.

| # Num | Methods | BL ↑ | CR↑ | Car ↑ | BA ↑ | AD ↑ | JA ↑ | Average ↑ |
|---|---|---|---|---|---|---|---|---|
| All | TabLLM (T0) | 0.761 | 0.737 | **0.845** | 0.906 | 0.864 | 0.691 | 0.801 |
| All | LIFT (GPT-3.5) | 0.689 | 0.691 | 0.729 | 0.825 | 0.810 | 0.647 | 0.732 |
| All | TP-BERTa (RoBERTa) | 0.761 | 0.730 | 0.826 | **0.916** | 0.844 | 0.659 | 0.789 |
| All | GTL (LLaMA2) | - | - | - | - | - | - | - |
| All | LIFT-ICL (GPT-4o) | - | - | - | - | - | - | - |
| All | P2T (GPT-4o) | - | - | - | - | - | - | - |
| All | FeatLLM (GPT-4o) | 0.768 | 0.701 | 0.589 | 0.887 | 0.842 | 0.540 | 0.721 |
| All | **DeLTa (GPT-4o) (Ours)** | **0.829** | **0.783** | 0.836 | 0.908 | **0.868** | **0.705** | **0.822** (↑ 2.6%) |
| 512 | TabLLM (T0) | 0.240 | 0.712 | 0.447 | 0.812 | 0.788 | 0.412 | 0.568 |
| 512 | LIFT (GPT-3.5) | 0.673 | 0.692 | 0.503 | 0.735 | 0.710 | 0.437 | 0.625 |
| 512 | TP-BERTa (RoBERTa) | 0.716 | 0.694 | 0.275 | **0.889** | 0.767 | 0.487 | 0.638 |
| 512 | GTL (LLaMA2) | - | - | - | - | - | - | - |
| 512 | LIFT-ICL (GPT-4o) | - | - | - | - | - | - | - |
| 512 | P2T (GPT-4o) | - | - | - | - | - | - | - |
| 512 | FeatLLM (GPT-4o) | 0.736 | 0.681 | 0.544 | 0.745 | 0.774 | 0.504 | 0.664 |
| 512 | **DeLTa (GPT-4o) (Ours)** | **0.748** | **0.719** | **0.738** | 0.799 | **0.790** | **0.522** | **0.719** (↑ 8.3%) |
| 256 | TabLLM (T0) | 0.344 | 0.690 | 0.471 | 0.785 | **0.787** | 0.373 | 0.575 |
| 256 | LIFT (GPT-3.5) | 0.651 | 0.682 | 0.431 | 0.724 | 0.773 | 0.398 | 0.610 |
| 256 | TP-BERTa (RoBERTa) | 0.721 | 0.677 | 0.408 | **0.853** | 0.772 | 0.464 | 0.649 |
| 256 | GTL (LLaMA2) | - | - | - | - | - | - | - |
| 256 | LIFT-ICL (GPT-4o) | - | - | - | - | - | - | - |
| 256 | P2T (GPT-4o) | - | - | - | - | - | - | - |
| 256 | FeatLLM (GPT-4o) | 0.729 | 0.658 | 0.459 | 0.740 | 0.772 | 0.473 | 0.639 |
| 256 | **DeLTa (GPT-4o) (Ours)** | **0.732** | **0.713** | **0.736** | 0.767 | 0.778 | **0.476** | **0.700** (↑ 7.9%) |
| 128 | TabLLM (T0) | 0.240 | **0.692** | 0.389 | 0.676 | 0.769 | 0.350 | 0.519 |
| 128 | LIFT (GPT-3.5) | 0.643 | 0.640 | 0.693 | 0.733 | 0.760 | 0.430 | 0.650 |
| 128 | TP-BERTa (RoBERTa) | 0.735 | 0.682 | 0.462 | **0.867** | 0.743 | 0.357 | 0.641 |
| 128 | GTL (LLaMA2) | - | - | - | - | - | - | - |
| 128 | LIFT-ICL (GPT-4o) | - | - | - | - | - | - | - |
| 128 | P2T (GPT-4o) | - | - | - | - | - | - | - |
| 128 | FeatLLM (GPT-4o) | 0.721 | 0.648 | 0.568 | 0.698 | 0.768 | **0.474** | 0.646 |
| 128 | **DeLTa (GPT-4o) (Ours)** | **0.746** | 0.683 | **0.722** | 0.749 | **0.774** | 0.457 | **0.688** (↑ 6.0%) |
| 64 | TabLLM (T0) | 0.245 | **0.673** | 0.386 | 0.602 | **0.785** | 0.268 | 0.493 |
| 64 | LIFT (GPT-3.5) | 0.643 | 0.643 | 0.500 | 0.728 | 0.748 | 0.430 | 0.615 |
| 64 | TP-BERTa (RoBERTa) | 0.648 | 0.646 | 0.419 | **0.840** | 0.761 | 0.414 | 0.621 |
| 64 | GTL (LLaMA2) | 0.613 | 0.638 | 0.512 | 0.719 | 0.750 | 0.414 | 0.608 |
| 64 | LIFT-ICL (GPT-4o) | 0.348 | 0.538 | 0.326 | 0.469 | 0.613 | 0.251 | 0.424 |
| 64 | P2T (GPT-4o) | 0.368 | 0.542 | 0.334 | 0.569 | 0.634 | 0.243 | 0.448 |
| 64 | FeatLLM (GPT-4o) | 0.697 | 0.643 | 0.508 | 0.723 | 0.749 | 0.450 | 0.628 |
| 64 | **DeLTa (GPT-4o) (Ours)** | **0.732** | 0.663 | **0.615** | 0.733 | 0.758 | **0.457** | **0.660** (↑ 5.0%) |
| 16 | TabLLM (T0) | 0.240 | **0.665** | 0.365 | 0.489 | 0.501 | 0.215 | 0.413 |
| 16 | LIFT (GPT-3.5) | 0.670 | 0.608 | 0.408 | 0.583 | 0.705 | 0.363 | 0.556 |
| 16 | TP-BERTa (RoBERTa) | 0.659 | 0.609 | 0.355 | 0.625 | 0.734 | 0.394 | 0.563 |
| 16 | GTL (LLaMA2) | 0.604 | 0.622 | 0.324 | 0.628 | **0.746** | 0.387 | 0.552 |
| 16 | LIFT-ICL (GPT-4o) | 0.457 | 0.492 | 0.324 | 0.308 | 0.512 | 0.198 | 0.382 |
| 16 | P2T (GPT-4o) | 0.226 | 0.471 | 0.320 | 0.105 | 0.499 | 0.203 | 0.304 |
| 16 | FeatLLM (GPT-4o) | 0.588 | 0.626 | **0.450** | 0.629 | 0.728 | **0.399** | 0.570 |
| 16 | **DeLTa (GPT-4o) (Ours)** | **0.682** | 0.638 | 0.369 | **0.674** | 0.730 | 0.379 | **0.579** (↑ 1.6%) |

Table 11: Comparison with LLM-based methods on all regression datasets ↓. We also calculate the average relative improvement of DeLTa against the best baseline method (↓ %). "# Num" indicates the number of training samples. This table serves as an extension of Table 9.

| # Num | Methods | CP↓ | CRR ↓ | CA ↓ | HO↓ | FR↓ | DI↓ | Average ↓ |
|---|---|---|---|---|---|---|---|---|
| All | TabLLM (T0) | - | - | - | - | - | - | - |
| All | LIFT (GPT-3.5) | 0.379 | 0.983 | 0.629 | 0.902 | 0.518 | 0.328 | 0.623 |
| All | TP-BERTa (RoBERTa) | 0.138 | 0.795 | 0.399 | 0.708 | 0.215 | 0.163 | 0.403 |
| All | GTL (LLaMA2) | - | - | - | - | - | - | - |
| All | LIFT-ICL (GPT-4o) | - | - | - | - | - | - | - |
| All | P2T (GPT-4o) | - | - | - | - | - | - | - |
| All | FeatLLM (GPT-4o) | - | - | - | - | - | - | - |
| All | **DeLTa (GPT-4o) (Ours)** | **0.116** | **0.780** | **0.351** | **0.529** | **0.200** | **0.130** | **0.351** (↓ 12.9%) |
| 512 | TabLLM (T0) | - | - | - | - | - | - | - |
| 512 | LIFT (GPT-3.5) | 0.432 | 1.167 | 0.752 | 1.385 | 0.589 | 0.368 | 0.782 |
| 512 | TP-BERTa (RoBERTa) | 0.186 | 0.837 | 0.528 | 0.760 | 0.383 | 0.280 | 0.496 |
| 512 | GTL (LLaMA2) | - | - | - | - | - | - | - |
| 512 | LIFT-ICL (GPT-4o) | - | - | - | - | - | - | - |
| 512 | P2T (GPT-4o) | - | - | - | - | - | - | - |
| 512 | FeatLLM (GPT-4o) | - | - | - | - | - | - | - |
| 512 | **DeLTa (GPT-4o) (Ours)** | **0.168** | **0.814** | **0.515** | **0.748** | **0.287** | **0.238** | **0.462** (↓ 6.9%) |
| 256 | TabLLM (T0) | - | - | - | - | - | - | - |
| 256 | LIFT (GPT-3.5) | 0.324 | 1.034 | 0.683 | 1.294 | 0.578 | 0.379 | 0.715 |
| 256 | TP-BERTa (RoBERTa) | 0.359 | 0.863 | **0.557** | **0.798** | 0.545 | 0.368 | 0.582 |
| 256 | GTL (LLaMA2) | - | - | - | - | - | - | - |
| 256 | LIFT-ICL (GPT-4o) | - | - | - | - | - | - | - |
| 256 | P2T (GPT-4o) | - | - | - | - | - | - | - |
| 256 | FeatLLM (GPT-4o) | - | - | - | - | - | - | - |
| 256 | **DeLTa (GPT-4o) (Ours)** | **0.234** | **0.841** | 0.568 | 0.799 | **0.397** | **0.277** | **0.519** (↓ 10.7%) |
| 128 | TabLLM (T0) | - | - | - | - | - | - | - |
| 128 | LIFT (GPT-3.5) | 0.290 | 1.070 | 0.710 | 1.300 | 0.660 | 0.430 | 0.743 |
| 128 | TP-BERTa (RoBERTa) | 0.431 | 0.903 | **0.597** | 0.880 | 0.654 | 0.390 | 0.642 |
| 128 | GTL (LLaMA2) | - | - | - | - | - | - | - |
| 128 | LIFT-ICL (GPT-4o) | - | - | - | - | - | - | - |
| 128 | P2T (GPT-4o) | - | - | - | - | - | - | - |
| 128 | FeatLLM (GPT-4o) | - | - | - | - | - | - | - |
| 128 | **DeLTa (GPT-4o) (Ours)** | **0.247** | **0.857** | 0.641 | **0.855** | **0.512** | **0.319** | **0.572** (↓ 11.0%) |
| 64 | TabLLM (T0) | - | - | - | - | - | - | - |
| 64 | LIFT (GPT-3.5) | 0.610 | 1.050 | 0.810 | 1.320 | 0.700 | 0.470 | 0.827 |
| 64 | TP-BERTa (RoBERTa) | 0.645 | 0.928 | 0.704 | 0.920 | 0.717 | 0.445 | 0.727 |
| 64 | GTL (LLaMA2) | 0.663 | 1.190 | 0.834 | 1.189 | 0.724 | 0.483 | 0.847 |
| 64 | LIFT-ICL (GPT-4o) | 0.637 | 1.213 | 0.820 | 1.045 | 0.710 | 0.494 | 0.820 |
| 64 | P2T (GPT-4o) | 0.697 | 1.335 | 0.868 | 1.452 | 0.745 | 0.513 | 0.935 |
| 64 | FeatLLM (GPT-4o) | - | - | - | - | - | - | - |
| 64 | **DeLTa (GPT-4o) (Ours)** | **0.382** | **0.888** | **0.701** | **0.919** | **0.634** | **0.383** | **0.651** (↓ 10.4%) |
| 16 | TabLLM (T0) | - | - | - | - | - | - | - |
| 16 | LIFT (GPT-3.5) | 0.890 | 1.130 | 1.010 | 1.350 | 0.920 | 0.670 | 0.995 |
| 16 | TP-BERTa (RoBERTa) | 0.814 | **0.941** | 0.924 | 1.008 | 0.969 | 0.669 | 0.887 |
| 16 | GTL (LLaMA2) | 0.804 | 1.184 | 0.904 | 1.258 | 0.854 | 0.624 | 0.938 |
| 16 | LIFT-ICL (GPT-4o) | 0.815 | 1.256 | 1.074 | 1.172 | 0.977 | 0.855 | 1.025 |
| 16 | P2T (GPT-4o) | 0.795 | 1.390 | 0.961 | 1.438 | 0.844 | 0.824 | 1.042 |
| 16 | FeatLLM (GPT-4o) | - | - | - | - | - | - | - |
| 16 | **DeLTa (GPT-4o) (Ours)** | **0.776** | 1.094 | **0.860** | **0.953** | **0.835** | **0.593** | **0.852** (↓ 4.0%) |

## A.8 Full comparison results with conventional baseline methods

We also provide the full results of DeLTa against non LLM-based baseline methods in full-data regimes in Table 12 and Table 13.

Table 12: Comparison with non LLM-based methods on all classification datasets ↑. We also calculate the average relative improvement of DeLTa against the best baseline method (↑ 3%).

| Methods | BL ↑ | CR↑ | Car ↑ | BA ↑ | AD ↑ | JA ↑ | Average ↑ |
|---|---|---|---|---|---|---|---|
| KNN | 0.7072 | 0.6429 | 0.7730 | 0.8903 | 0.8186 | 0.5168 | 0.7248 |
| CART | 0.7871 | 0.6829 | 0.7253 | 0.8900 | 0.8392 | 0.5642 | 0.7481 |
| MLP | 0.6882 | 0.7171 | 0.6974 | 0.9013 | 0.8523 | 0.7043 | 0.7601 |
| RandomForest | 0.8061 | 0.7371 | 0.7944 | 0.8915 | 0.8589 | 0.6588 | 0.7911 |
| XGBoost | 0.7605 | 0.7171 | 0.6102 | 0.9051 | 0.8709 | 0.7140 | 0.7630 |
| Catboost | 0.7757 | 0.7543 | 0.6036 | 0.9080 | **0.8726** | 0.7217 | 0.7727 |
| ResNet | 0.6464 | 0.7200 | 0.6332 | 0.9020 | 0.8512 | 0.7134 | 0.7444 |
| FT-Transformer | 0.7871 | 0.7229 | 0.7862 | 0.9030 | 0.8590 | **0.7272** | 0.7976 |
| SAINT | 0.7490 | 0.7486 | 0.7582 | 0.9051 | 0.8560 | 0.7050 | 0.7870 |
| TabPFN (Version 2) | 0.7985 | 0.7286 | 0.4507 | **0.9095** | 0.8611 | 0.7020 | 0.7417 |
| ModernNCA | 0.8099 | 0.7286 | 0.7368 | 0.9046 | 0.8702 | 0.7248 | 0.7958 |
| **DeLTa (Ours)** | **0.8289** | **0.7829** | **0.8355** | 0.9080 | 0.8677 | 0.7048 | **0.8213** (↑ 3%) |

Table 13: Comparison with non LLM-based methods on all regression datasets ↓. We also calculate the average relative improvement of DeLTa against the best baseline method (↓ 2.4%).

| Methods | CP↓ | CRR ↓ | CA ↓ | HO↓ | FR↓ | DI↓ | Average ↓ |
|---|---|---|---|---|---|---|---|
| KNN | 0.2347 | 1.0975 | 0.5893 | 0.7970 | 0.4628 | 0.2135 | 0.5658 |
| CART | 0.2533 | 0.8196 | 0.7100 | 0.8433 | 0.6462 | 0.3504 | 0.6038 |
| MLP | 0.1318 | 0.8244 | 0.4628 | 0.5949 | 0.2072 | 0.3068 | 0.4213 |
| RandomForest | 0.1521 | 0.8093 | 0.5057 | 0.6692 | 0.3712 | 0.1626 | 0.4450 |
| XGBoost | 0.1285 | 0.7846 | 0.4037 | 0.6169 | 0.2291 | 0.1379 | 0.3835 |
| Catboost | 0.1210 | **0.7779** | 0.3723 | 0.5672 | 0.2024 | 0.1306 | 0.3619 |
| ResNet | 0.1362 | 0.8194 | 0.4434 | 0.5931 | 0.2083 | 0.2143 | 0.4025 |
| FT-Transformer | 0.1187 | 0.7822 | 0.3991 | 0.5851 | 0.2067 | 0.1680 | 0.3766 |
| SAINT | 0.1371 | 0.8276 | 0.4385 | 0.6007 | 0.2063 | 0.1396 | 0.3916 |
| TabPFN (Version 2) | 0.1375 | 0.7956 | **0.3440** | 0.5527 | 0.2001 | 0.1282 | 0.3597 |
| ModernNCA | 0.1242 | 0.7876 | 0.3656 | 0.6270 | 0.2027 | **0.1280** | 0.3725 |
| **DeLTa (Ours)** | **0.1156** | 0.7804 | 0.3507 | **0.5293** | **0.2000** | 0.1303 | **0.3511** (↓ 2.4%) |

## A.9 Computational efficiency

Table 14: Runtime in seconds of DeLTa and other LLM-based baseline methods for the training and inference phase, conducted on Adult dataset. "# Num" indicates number of training samples. †These methods require querying LLMs for each test sample. * These methods do not require querying LLMs at inference phase.

| # Num | Stage | TabLLM† | LIFT† | TP-BERTa† | GTL† | LIFT-ICL† | P2T† | FeatLLM* | **DeLTa\* (Ours)** |
|---|---|---|---|---|---|---|---|---|---|
| All | Train | 177371.37 | 153689.14 | 1319.35 | N/A | N/A | N/A | 1231.22 | **35.20** |
| | Inference | 179.21 | 90149.48 | 1.58 | N/A | N/A | N/A | 0.14 | **0.09** |
| 64 | Train | 288.61 | 191.75 | 316.40 | N/A | N/A | N/A | 1047.55 | **23.09** |
| | Inference | 170.42 | 89684.76 | 5.01 | 153501.02 | 176726.13 | 200258.24 | 0.14 | **0.08** |

### A.10 Ablation results

Let $F(x)$ denote the well-trained ensemble of decision trees (Random Forest), and $\mathcal{R}$ the corresponding decision tree rule set. DeLTa first performs a rule refinement procedure via LLM to derive a new rule $r^*$, and subsequently applies an error correction mechanism to compute a sample-specific error correction vector based on $r^*$, which is then added to $F(x)$ to produce the final prediction. We further conduct ablation study to demonstrate the effectiveness of key components of DeLTa, as illustrated in Table 15 and Table 16, by comparing DeLTa against different groups of variants.

- *Variant A to B do not use the rule refinement process or the error correction process:* (1) variant A, that is identical to the single decision tree (CART); (2) variant B, that is identical to the existing ensemble of decision trees $F(x)$ (Random Forest).

- *Variant C to D incorporate the rule refinement process and utilize the $r^*$ to predict labels, while removes the error correction process:* (3) variant C, that treats the $r^*$ as a standalone decision tree by assigning the input $x$ to a specific leaf node; (4) variant D, that further appends $r^*$ to the existing ensemble of decision trees $F(x)$.

- *Variant E includes both the rule refinement process and the error correction process, and utilizes the $r^*$ to predict both labels and error correction vectors:* (5) variant E, that uses $r^*$ to predict labels like variant D, and also uses $r^*$ to predict error correction vectors like DeLTa. The final output is the sum of the predicted label and the predicted error correction vector.

- *Variant F removes the rule refinement process, but remains the error correction process:* (6) variant F, that replaces the LLM generated $r^*$ with $r \in \mathcal{R}$ derived from Random Forest to predict the error correction vector, and then adds it to the existing ensemble of decision trees $F(x)$ as the final prediction.

Table 15: Analysis on the effects of different components of DeLTa on classification tasks. "RF" corresponds to Random Forest.

| Variants | Rule Refinement | Need RF's output | Error Correction | BL ↑ | CR↑ | Car ↑ | BA ↑ | AD ↑ | JA ↑ | Average ↑ |
|---|---|---|---|---|---|---|---|---|---|---|
| A (CART) | – | – | – | 0.787 | 0.683 | 0.725 | 0.890 | 0.839 | 0.564 | 0.748 |
| B (RF) | – | ✓ | – | 0.806 | 0.737 | 0.794 | 0.892 | 0.859 | 0.659 | 0.791 |
| C | ✓ (for label) | – | – | 0.811 | 0.766 | 0.818 | 0.900 | 0.853 | 0.677 | 0.804 |
| D | ✓ (for label) | ✓ | – | 0.812 | 0.771 | 0.819 | 0.899 | 0.856 | 0.671 | 0.805 |
| E | ✓ (for residual and label) | ✓ | ✓ | 0.816 | 0.779 | 0.829 | 0.895 | 0.858 | 0.679 | 0.809 |
| F | – | – | ✓ | 0.810 | 0.726 | 0.778 | 0.892 | 0.842 | 0.613 | 0.777 |
| **DeLTa (Ours)** | ✓ (for residual) | ✓ | ✓ | **0.829** | **0.783** | **0.836** | **0.908** | **0.868** | **0.705** | **0.821** |

Table 16: Analysis on the effects of different components of DeLTa on regression tasks. "RF" corresponds to Random Forest.

| Variants | Rule Refinement | Need RF's output | Error Correction | CP↓ | CRR ↓ | CA ↓ | HO↓ | FR↓ | DI↓ | Average ↓ |
|---|---|---|---|---|---|---|---|---|---|---|
| A (CART) | – | – | – | 0.253 | 0.820 | 0.710 | 0.843 | 0.646 | 0.350 | 0.604 |
| B (RF) | – | ✓ | – | 0.152 | 0.809 | 0.506 | 0.669 | 0.371 | 0.163 | 0.445 |
| C | ✓ (for label) | – | – | 0.136 | 0.808 | 0.402 | 0.607 | 0.272 | 0.139 | 0.394 |
| D | ✓ (for label) | ✓ | – | 0.144 | 0.800 | 0.387 | 0.583 | 0.258 | 0.138 | 0.385 |
| E | ✓ (for residual and label) | ✓ | ✓ | 0.125 | 0.799 | 0.379 | 0.552 | 0.251 | 0.142 | 0.375 |
| F | – | – | ✓ | 0.218 | 0.819 | 0.544 | 0.811 | 0.619 | 0.324 | 0.556 |
| **DeLTa (Ours)** | ✓ (for residual) | ✓ | ✓ | **0.116** | **0.780** | **0.351** | **0.529** | **0.200** | **0.130** | **0.351** |

In the Table 4 in the original paper, DeLTa w/o "RR" corresponds to the variant F, and DeLTa w/o "EC" corresponds to the variant C. Here, "RR" denotes the rule refinement process, and "EC" denotes the error correction process. The results show that the decision tree rule refinement via LLM is important. Directly using the refined rule $r^*$ to predict labels could also enhance the performance (variant C and D). And the error correction strategy could further enhance tabular prediction performance (variant E and DeLTa). However, without $r^*$, the effect of error correction is limited (variant F). Therefore, the removal of any of the components degrades the performance of DeLTa.

## A.11 Hyper parameter sensitivity analysis

We incorporate the sensitivity analysis for the number of LLM querying times for DeLTa in Fig. 7. It is reasonable for us to set the querying times to 10 by default.

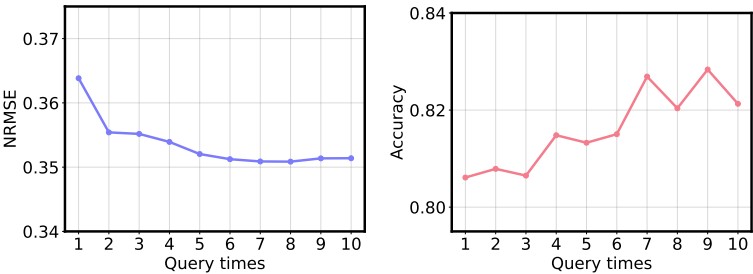

Figure 7: The effect of the number of LLM queries on performance averaged over all datasets.

## A.12 Additional analysis

Fig. 8 shows the label predictions of ours: $F(x) + \Delta_x$ and ours w/o $\Delta_x$: $F(x)$. We can observe that two categories of samples are overlapped in some regions. $F(x)$ struggles to distinguish them, but DeLTa could correct the prediction errors of $F(x)$ to enhance the prediction for such complex data patterns.

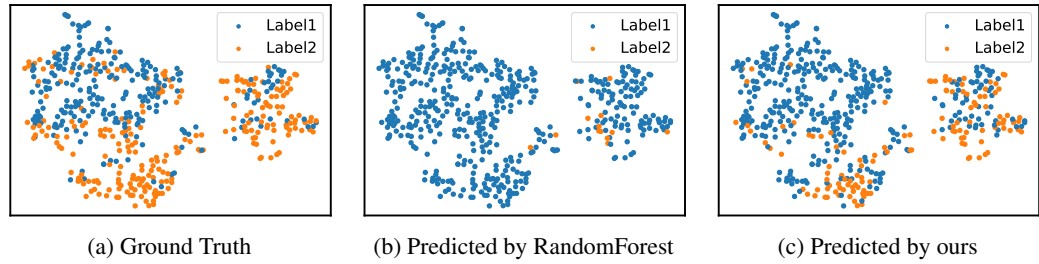

(a) Ground Truth                (b) Predicted by RandomForest                (c) Predicted by ours

Figure 8: T-SNE visualization of label prediction of DeLTa w/ and w/o error correction vector $\Delta_x$ (i.e., Random Forest) on BA dataset.

We compute the intra-node sample distance averaged over all leaf nodes partitioned by refined rule $r^*$ and random forest rules, respectively. The results in Table 17 show that LLM could generate a better rule, where samples grouped within the same leaf node exhibit greater statistical similarity.

Table 17: Average intra-node distance comparison.

|  | BL | Credit | Car | Bank | AD | JA | Average |
|---|---|---|---|---|---|---|---|
| RandomForest Rule | 2.8520 | 6.0938 | 2.6609 | 5.0186 | 4.7781 | 9.9177 | 5.2202 |
| **Refined Rule** | **1.3551** | **5.0503** | **2.5059** | **4.7078** | **4.2950** | **9.4011** | **4.5525** (↓ 12.79%) |

We also provide an example of the original decision tree rule set derived from Random Forest in Fig. 9, along with the corresponding refined rule generated by the LLM in Fig. 10. This example is taken from the low-data regime with 16 training samples on the Credit dataset.

```
Tree 0 rules:
|--- feature_2 <= 0.05
|   |--- class: 1.0
|--- feature_2 > 0.05
|   |--- feature_8 <= -0.26
|   |   |--- feature_10 <= -1.05
|   |   |   |--- class: 0.0
|   |   |--- feature_10 > -1.05
|   |   |   |--- class: 1.0
|   |--- feature_8 > -0.26
|   |   |--- class: 0.0

-----------------------------
Tree 1 rules:
|--- feature_16 <= 0.82
|   |--- feature_15 <= -0.99
|   |   |--- class: 0.0
|   |--- feature_15 > -0.99
|   |   |--- feature_5 <= 0.20
|   |   |   |--- class: 1.0
|   |   |--- feature_5 > 0.20
|   |   |   |--- feature_18 <= 0.18
|   |   |   |   |--- feature_10 <= -1.05
|   |   |   |   |   |--- class: 0.0
|   |   |   |   |--- feature_10 > -1.05
|   |   |   |   |   |--- class: 1.0
|   |   |   |--- feature_18 > 0.18
|   |   |   |   |--- class: 1.0
|--- feature_16 > 0.82
|   |--- feature_4 <= -0.91
|   |   |--- class: 1.0
|   |--- feature_4 > -0.91
|   |   |--- class: 0.0

-----------------------------
Tree 2 rules:
|--- feature_2 <= 0.05
|   |--- class: 1.0
|--- feature_2 > 0.05
|   |--- feature_17 <= 0.82
|   |   |--- class: 0.0
|   |--- feature_17 > 0.82
|   |   |--- class: 1.0

-----------------------------
```

Figure 9: An example of three tree rules obtained by the Random Forest.

To design a more effective decision rule based on the given CART tree rules and enhance performance, we should aim to combine the features that appear frequently and are pivotal in making decisions, while also ensuring that the rule structure is concise and balanced.

From the provided rules, features like `feature_2`, `feature_10`, `feature_16`, and `feature_17` appear to play significant roles in decision splits. By analyzing the tree depth and split thresholds, we can refine the rules.

Here's an improved rule set in the specified format, incorporating insights from the provided trees:

```python
self.tree = {
    "feature": 2,
    "threshold": 0.05,
    "operator": "<=",
    "left": {"id": "leaf_1"},
    "right": {
        "feature": 16,
        "threshold": 0.82,
        "operator": "<=",
        "left": {
            "feature": 10,
            "threshold": -1.05,
            "operator": "<=",
            "left": {"id": "leaf_2"},
            "right": {
                "feature": 18,
                "threshold": 0.18,
                "operator": ">",
                "left": {"id": "leaf_3"},
                "right": {"id": "leaf_4"},
            },
        },
        "right": {
            "feature": 17,
            "threshold": 0.82,
            "operator": "<=",
            "left": {"id": "leaf_5"},
            "right": {"id": "leaf_6"},
        },
    },
}
```

### Explanation:

1. **Choosing Features and Thresholds**:
   - Feature `2` is a critical deciding factor across multiple trees, with a consistent threshold of `0.05`.
   - Feature `16` was chosen due to its significance in Tree 1 with a prominent threshold of `0.82`.
   - Feature `10` serves as a key deciding factor for certain conditions, particularly effective at distinguishing classes when less than `-1.05`.
   - Features `17` and `18` have been adapted for further splits on the right branch, consistent with their impactful splits in Tree 2 and Tree 1, respectively.

2. **Structure**:
   - The initial split on `feature_2` immediately classifies a region into leaf 1.
   - The decision structure on the right branch further splits using `feature_16`, guiding to other regions depending on values of `feature_10`, `feature_18`, and `feature_17`.

This refined rule set strategically uses key features to balance simplicity and depth, aiming for improved decision-making accuracy and compactness in the decision space division.

Figure 10: An example of LLM's response for rule refinement.

**How an LLM refines decision tree rules.**

*(i) The intuition.* Traditional methods like CART use information gain to make a series of locally optimal decisions, allowing a data-driven algorithm to find the best feature splits. This can lead to rules that are brittle or overfit to the specific data sample used for that tree. DeLTa, instead, utilizes the LLM to analyze diverse rules from a Random Forest and performs a logic-based reasoning task: (a) It recognizes patterns across multiple "expert" rules to identify features and decision paths that are consistently important. (b) It abstracts the core logic splits from individual trees. (c) It synthesizes a new, globally coherent rule that represents a consensus of the most robust patterns.

*(ii) A concrete example of a better rule.* As shown in Fig. 9 and Fig. 10, we provide a set of decision tree rules from Random Forest and refined rule by the LLM respectively. This is from the Credit dataset, where each sample (person) is classified as good or bad credit risks according to the set of features. To provide deeper insight into how the refined rule improves upon the original ones, we also provide the average SHAP values [70] of features under the Random Forest and our method respectively in Table 18. SHAP assigns each feature an importance value (higher is more important) for a particular prediction, thus it serves an unified framework for interpreting predictions.

The results show that, after LLM refinement, the SHAP importance of features such as "credit_history" (feature_16) and "employment" (feature_10) are increased, which aligns with the financial expertise. Let us take a closer look into Fig. 9 and Fig. 10, for "credit_history" (feature_16), this feature appears near the top of the second tree, indicating its potential significance. However, only 1 out of 3 decision trees in the original Random Forest rule set explicitly splits on this feature, which may dilute its influence in the ensemble prediction. For "employment" (feature_10), although it appears in 2 out of 3 rules, its splits occur late in the tree, leading to a low overall contribution in prediction. In contrast, the LLM-refined rule elevates the prominence of these features.

*(iii) Why do traditional techniques not capture such a rule?* More broadly, a single decision tree with information gain has the risk of overfitting. Although Random Forest ensembles the outputs from all tree rules, the inherent relationships and interactions among these rules are ignored. As the number of trees increases, analyzing these independent rules is gradually becoming more and more difficult, let alone utilizing these rules to partition feature space. To this end, we propose to leverage LLMs to analyze and summarize these rules into a refined rule due to the powerful logical reasoning ability of LLM. This enables the generation of improved partitioning strategies over the feature space, thereby promoting statistical coherence among samples assigned to the same leaf node. To verify it, we compute the intra-node sample distance over all leaf nodes partitioned by LLM refined rule $r^*$, and observe that the distance of $r^*$ is lower than original rule $r \in \mathcal{R}$ from Random Forest, as illustrated in Fig. 2 and Table 17.

Table 18: SHAP values of different features.

| Original | property_magnitude 0.1487 | other_payment_plans 0.0471 | checking_status 0.0410 | purpose 0.0355 | housing 0.0326 | foreign_worker 0.0251 | job 0.0213 | credit_history 0.0189 | savings_status 0.0105 | personal_status 0.0070 |
|---|---|---|---|---|---|---|---|---|---|---|
| **Ours** | property_magnitude 0.1055 | credit_history 0.0436 | checking_status 0.0342 | purpose 0.0174 | employment 0.0155 | other_payment_plans 0.0144 | foreign_worker 0.0132 | housing 0.0106 | savings_status 0.0096 | job 0.0095 |

**Assessing the inherent quality of LLM-rewritten rules.** We assess this in a principled way from three perspectives: (i) SHAP value analysis, (ii) intra-node distance comparison and (iii) the transparency of the LLM's process, where the LLM provides a human-readable rationale for its actions, which allows us to inspect and understand its reasoning process.

**Comparison on additional datasets.** We have include additional datasets from the WhyTrees [11] benchmark. Specifically, we selected a subset of representative datasets that span different scales and feature characteristics, including the large-scale, high-dimensional "Year" dataset, to evaluate DeLTa's scalability and generalizability. The updated results, summarized in Table 19 and Table 20, further demonstrate DeLTa's strong and consistent performance across a more diverse and standardized set of tabular tasks.

Table 19: Additional tabular dataset properties. "#objects" indicates the number of samples in the dataset. "#num. features" indicates the number of numerical features, and "#cat. features" indicates the number of categorical features.

| Dataset | #objects | #num. features | #cat. features | metric | #classes |
|---|---|---|---|---|---|
| rl | 4970 | 5 | 7 | Acc. | 2 |
| GiveMeSomeCredit | 16714 | 10 | 0 | Acc. | 2 |
| Year | 515345 | 90 | 0 | NRMSE | – |

Table 20: Comparison with baseline methods on additional datasets. "(L)" and "(T)" indicates correspond to LLM-based and traditional methods, respectively.

| Dataset | TabLLM (L) | FeatLLM (L) | LIFT (L) | TP-BERTa (L) | KNN (T) | CART (T) | MLP (T) | RandomForest (T) | XGBoost (T) | CatBoost (T) | FT-Transformer (T) | **DeLTa** (L) |
|---|---|---|---|---|---|---|---|---|---|---|---|---|
| rl ↑ | 0.7716 | 0.7243 | 0.6349 | 0.7394 | 0.6258 | 0.6308 | 0.6630 | 0.7022 | 0.7726 | 0.7746 | 0.7072 | **0.7897** |
| GiveMeSomeCredit ↑ | 0.7765 | 0.7137 | 0.6534 | 0.7810 | 0.6063 | 0.7514 | 0.7469 | 0.7768 | 0.7789 | 0.7792 | 0.7610 | **0.7816** |
| Year ↓ | - | - | 1.0540 | 0.8258 | 1.0459 | 0.9374 | 0.8134 | 0.8547 | 0.8376 | 0.8157 | 0.8090 | **0.8015** |

### A.13 Discussion

The existing LLM-based methods for tabular prediction suffer from two key inherent issues: (i) data perspective: existing data serialization methods lack universal applicability for structured tabular data, and may pose privacy risks through direct textual exposure, and (ii) model perspective: LLM fine-tuning methods struggle with tabular data, and in-context learning scalability is bottle-necked by input length constraints (suitable for few-shot learning). The proposed DeLTa could well solve the aforementioned challenges: *Solving the data issue:* Unlike serialization methods that convert each sample into unnatural text formats, decision tree rules are composed of simple comparisons between feature values and thresholds, forming logical, interpretable structures that can be naturally expressed in text without relying on semantic columns names. In addition, decision tree rules represent global feature space partitioning rule rather than individual samples, which helps mitigate privacy concerns by avoiding exposure of sample-level information. *Solving the model issue:* Moreover, the powerful reasoning ability of LLMs can be leveraged to redesign decision tree rules and help trees with aggregating their decisions, rather than directly using LLMs to generate label predictions. Notably, DeLTa avoids serializing tabular data into natural language format, and does not require additional domain-specific expertise or semantic information, such as explicit feature names and detailed task background knowledge. Furthermore, DeLTa can be applied in full data learning setting without LLM fine-tuning.

Overall, DeLTa achieves the highest average performance in all settings, including classification and regression, in both full and low-data regimes. In addition, DeLTa is computationally efficient in both training and inference stages. This efficiency stems from key properties: (i) DeLTa utilizes the reason ability of LLM without fine-tuning LLM, (ii) DeLTa avoids querying LLMs to generate predictions for individual samples. Instead, it only needs to query LLM via API to generate one refined decision tree rule for one dataset, enabling significantly more efficient use of LLM resources.

### A.14 Limitation

Although the primary focus of this work is tabular data, the flexibility of DeLTa opens up exciting possibilities for extensions. For example, DeLTa can be extended to other domains such as time series, graph and so on. We leave the extensions of DeLTa as a future work. While DeLTa avoids tabular data serialization, it requires training decision trees and extracting corresponding decision rules. However, single decision tree may be prone to overfitting, potentially resulting in suboptimal rules and limited performance improvements in our framework. To mitigate this issue, we adopt Random Forest to generate a diverse set of decision tree rules. This strategy is not only effective but also easy to implement using standard libraries such as scikit-learn, which conveniently supports both model training and rule extraction.

Another limitation of DeLTa is that its performance advantage over non-LLM baselines narrows with increasing dataset sizes. Upon deeper analysis, we have found that DeLTa's performance advantage is more pronounced when the dataset size is relatively small, compared to non-LLM baselines. Conversely, the performance advantage of data-intensive baseline methods, such as neural networks, tends to increase with larger dataset sizes. This pattern may be attributed to two main factors: (i) **Nature of Base Rules**: Our method refines the base rules extracted from Random Forest (RF). While RF is effective due to its ensemble nature, its trees are independently trained without joint optimization. As the data volume increases, this independent training may limit RF's ability to fully leverage large-scale data. Consequently, the quality of the base rules may plateau with increasing data, which also affects the upper bound of the refined rules. (ii) **Comparison to Neural Networks (NNs)**: In contrast, neural networks are typically data-intensive models that require large amounts of training data to perform well and generalize effectively. Their performance tends to scale better with larger datasets, as they can learn complex feature interactions.

### A.15 Future direction

A key direction involves developing more scalable base rule extraction mechanisms. Exploring ensembles of jointly optimized trees or hybrid models that better leverage large-scale data could raise the performance ceiling for refinement. Furthermore, integrating the rule-based paradigm with data-intensive architectures—for instance, by using neural networks to generate or pre-select

candidate rules, or by designing LLM-driven fine-tuning that adapts to dataset scale—presents a promising avenue to combine the interpretability of rules with the scalability of neural models.

## A.16 Broader impacts

DeLTa advances the integration of LLMs with structured tabular data by introducing a novel way that leverages logical decision tree rules as intermediaries. By representing the entire dataset through decision tree rules, DeLTa avoids the need for sample-level serialization. This makes it particularly well-suited for structured tabular data with heterogeneous features, in contrast to the unstructured nature of textual data. DeLTa also offers practical benefits in terms of data privacy. It could help mitigate privacy concerns by avoiding exposure of sample-level information, which is useful in privacy-critical domains such as healthcare and finance. Furthermore, DeLTa can be applied in full data learning setting without LLM fine-tuning, thereby offering a cost-efficient and scalable solution for integrating LLMs into tabular learning tasks.

