# OpenReview forum: "LLM Meeting Decision Trees on Tabular Data"
_NeurIPS.cc/2025/Conference — NeurIPS 2025 spotlight_

### Official Review · Reviewer_WTPj · 2025-06-30

**Clarity:** 3
**Significance:** 3
**Originality:** 3
**Rating:** 5
**Confidence:** 3

**Summary:**

This paper introduces DeLTa (Decision Tree Enhancer with LLM-derived Rule for Tabular Prediction), a novel approach that integrates Large Language Models (LLMs) with decision trees to improve predictions on tabular data. According to authors, DeLTa addresses three key issues in prior LLM-based methods for tabular data:
1) lack of universal applicability and privacy risks from data serialization usually used in current LLM-based tabular approaches.
2) challenges of LLM fine-tuning
3) scalability limitations in-context learning

This approach seems to be using logical decision tree rules as intermediaries and it leverages the reasoning ability of LLMs to generate improved rules from existing decision tree rule sets without requiring LLM fine-tuning.

**Questions:**

- The rule r* may contain threshold or other information depending on the feature types. Does this leak privacy in some form?
- Apart from the experimental results, which are showing good performance, how can we measure if the LLM rewriting rules (or correcting them) is performing well? In a way, this is also a bit of a zero-shot task for the LLM itself.
- How is η set? what's the interpretation of it in the context of DeLta?
- Is it fair to say that the method always assumes that the LLM will always find a r* leading to lower loss but this is an assumption?

**Ethical Concerns:**

["NO or VERY MINOR ethics concerns only"]

**Final Justification:**

That's great. Thanks

**Limitations:**

Maybe clarify:
- Privacy leakage beyond samples
- Discuss when LLM fails to improve rules

**Paper Formatting Concerns:**

No concerns

**Quality:**

3

**Strengths And Weaknesses:**

The authors seem to deliver on their promise to avoid serialization, fine tuning and scalability. The main advantage of the paper seems to rely on improved rules from existing decision tree rule sets and using these rules to calibrate predictions. The method achieves state-of-the-art performance on diverse tabular benchmarks in both full-data and low-data regimes.

Very interesting approach to combine 2 steps to refine tree rules and training a gradient net to correct errors.

My main comment is that this seems more of decision tree enhancer system that introduces advantages natural to LLMs rather than a LLM-based solution for tabular data problems. And that's fine, the way the authors use an LLM to synthesize a refined rule r* from multiple tree rules, and then apply that to structure a residual correction layer, reminds me of boosting. Would you say DeLTa is a boosting-inspired framework but the LLM refines instead of being a data-driven iterative fitting?

Maybe also citing TabNet (it's one of the most well-known deep learning models for tabular data with attention and interpretability), or TabTransformer (pioneered using transformers on column embeddings).

Typos:
- throughough
- an domain

---

> ### Author Rebuttal · Authors · 2025-07-31
>
> R to W1: Thanks for your thoughtful comment. You are right that DeLTa shares some conceptual resemblance with boosting.
> Traditional Boosting would train a new, simple weak learner (e.g., a shallow decision tree) on the residuals, allowing a data-driven algorithm to find the best feature splits. DeLTa, instead, utilizes the LLM to analyze diverse rules and then synthesize a feature space partitioning rule, $r^*$. This rule then structures the error-correction stage. The LLM's role is not a greedy search on data but a logical synthesis of existing rules.
> While DeLTa undoubtedly benefits decision trees, its essence is to explore a novel use of LLMs for tabular data, which fundamentally differs from existing LLM-based tabular prediction paradigms.
>
> Most current LLM-based approaches typically first serialize tabular data into natural language descriptions, and then tune LLMs or directly infer on these serialized data. There are two key limitations: (1) it is challenging to design such suitable template due to the modality gap between unstructured text and structured tabular data. And it may also pose privacy risks through direct textual exposure, and (2) LLM fine-tuning methods struggle with tabular data, and in-context learning scalability is bottle-necked by input length constraints (suitable for few-shot learning).
>
> In contrast, DeLTa integrates LLM into tabular data via logical decision tree rules as intermediaries: the LLM is not used to predict sample-level outputs directly, but instead to generate a refined decision rule by understanding multiple tree rules.
> This rule-level interface has several distinct advantages: it (a) avoids serialize tabular data into natural language format, (b) preserves data privacy and does not require domain-specific information, and (c) can be applied in full data learning setting without LLM-finetuning.
> In essence, while the error-correction mechanism is boosting-inspired, the core novelty is using an LLM for logical rule refinement, establishing a new and more robust way to bridge the gap between LLMs and structured data.
>
> R to W2: Thanks for your comment. We will cite these papers in our main manuscript.
>
> R to W3: Thanks for pointing this out. We will carefully revised the corresponding parts of manuscript to make it more clear.
>
> R to Q1: Thanks for your comment. As discussed in Appendix A.4, including the prompt example in Fig. 6,  although we need to submit the rules to LLM, we do not need to upload additional domain-specific semantic information when designing prompt, such as explicit feature names or detailed task background knowledge. While the decision tree rules may contain feature threshold, all features fed into LLM are anonymized. Moreover, decision tree rules represent global feature space partitioning rule rather than individual samples. That is to say, the LLM operates only on abstracted, symbolic representations that reflect model logic rather than raw data. Moving beyond some LLM-based tabular learning methods that expose the serialized tabular samples containing raw feature values to LLMs, ours significantly mitigate privacy concerns.
>
> R to Q2:  Thanks for your comment. We agree that evaluating the quality of the rule rewriting process itself, independent of final task performance, is crucial for understanding why our method is effective. We assess this in a principled way from three perspectives: (i) SHAP value analysis, (ii) intra-node distance comparison and (iii) the transparency of the LLM's process.
>
> **(i) SHAP value analysis.** As shown in Fig. 10 and Fig. 11 of Appendix, we provide a set of decision tree rules from Random Forest and refined rule by the LLM respectively. This is from the Credit dataset, where each sample (person) is classified as good or bad credit risks according to the set of features. To provide deeper insight into how the refined rule improves upon the original ones, we also provide the average SHAP values [1] of features under the Random Forest and our method respectively as follows. SHAP assigns each feature an importance value (higher is more important) for a particular prediction, thus it serves an unified framework for interpreting predictions.
>
> The results show that, after LLM refinement, the SHAP importance of features such as ''credit\_history'' (feature\_16) and ''employment'' (feature\_10) are increased, which aligns with the financial expertise. Let us take a closer look into Fig. 10 and Fig. 11 of Appendix, for ''credit\_history'' (feature\_16), this feature appears near the top of the second tree, indicating its potential significance. However, only 1 out of 3 decision trees in the original Random Forest rule set explicitly splits on this feature, which may dilute its influence in the ensemble of Random Forest prediction.
> For ''employment'' (feature\_10), although it appears in 2 out of 3 rules, its splits occur late in the tree, leading to a low overall contribution in the ensemble of Random Forest prediction.
> In contrast, the LLM-refined rule elevates the prominence of these features.
> |Original feature|Importance|Our feature|Importance|
> |-|-|-|-|
> |property\_magnitude|0.1487|property\_magnitude|0.1055|
> |other\_payment\_plans|0.0471|credit\_history|0.0436|
> |checking\_status|0.0410|checking\_status|0.0342|
> |purpose|0.0355|purpose|0.0174|
> |housing|0.0326|employment|0.0155|
> |foreign\_worker|0.0251|other\_payment\_plans|0.0144|
> |job|0.0213|foreign\_worker|0.0132|
> |credit\_history|0.0189|housing|0.0106|
> |savings\_status|0.0105|savings\_status|0.0096|
> |personal\_status|0.0070|job|0.0095|
>
> [1] A Unified Approach to Interpreting Model Predictions. NeurIPS 2017.
>
> **(ii) Intra-node distance comparison.** Although Random Forest ensembles the outputs from all tree rules, the inherent relationships and interactions among these rules are ignored. As the number of trees increases, analyzing these independent rules is gradually becoming more and more difficult, let alone utilizing these rules to partition feature space. To this end, we propose to leverage LLMs to analyze and summarize these rules into a refined rule due to the powerful logical reasoning ability of LLM. This enables the generation of improved partitioning strategies over the feature space, thereby promoting statistical coherence among samples assigned to the same leaf node.
> To verify it, we compute the intra-node sample distance over all leaf nodes partitioned by LLM refined rule $r^* $, and observe that the distance of $r^*$ is lower than original rule $r\in \mathcal{R}$ from Random Forest, as illustrated in Fig. 2 of original paper and Table 17 of Appendix.
>
> **(iii) Transparency of the LLM's Process: The Self-Explanation.** Finally, our method is not a complete black box. The LLM provides a human-readable rationale for its actions, which allows us to inspect and understand its reasoning process. As shown in Fig. 11 of Appendix, the LLM generates a detailed "Explanation" for its refined rule. It explicitly breaks down its logic into two parts: "Choosing Features and Thresholds" and "Structure". It articulates why feature\_2 was chosen (as a "critical deciding factor across multiple trees" ) and how it constructed the new tree (the "initial split on feature\_2 immediately classifies a region into leaf 1" ).
>
> In summary, we evaluate the LLM’s rule-rewriting performance through a multi-faceted approach. Together, these methods provide a robust and principled way to measure the success of this core component of our work. We will revise our manuscript to clarify it.
>
> R to Q3: $\eta$ is set to 0.2 by default. In Eq. 6 of original paper, we need to move prediction produced by the Random Forest $F(x)$ in the direction of negative gradient by one step, and $\eta\in \mathbb{R}_+$ is the hyperparameter controlling the step size. In this context, $\eta$ plays a similar role to the learning rate in gradient descent, and it determines the strength of the correction. We found that $\eta=0.2$ works robustly.
>
> R to Q4: Thanks for this insightful observation. You are correct that DeLTa relies on the LLM's ability to generate a useful rule $r^* $. However, we do not assume this process is deterministically perfect. Notably, DeLTa is designed with two specific mechanisms that make it highly robust to occasional, suboptimal generations from the LLM. First, we mitigate the risk of a single poor LLM output by querying the LLM 10 times by default and aggregating the results, as discussed in Appendix (Line 558). This helps to reduce variance and smooth out occasional underperforming generations. We have indeed observed rare cases (e.g., on the Adult 64-shot setting) where the LLM failed to synthesize a novel rule and instead largely replicated an input tree. Our aggregation strategy is designed to minimize the impact of such infrequent events. Second, and more importantly, the error correction (EC) stage of DeLTa acts as a powerful safety net, ensuring a strong final performance even if the initial LLM-generated rule is imperfect. The LLM refined rule $r^* $ is not the final predictor. Its primary role is to structure the problem for Gradient net, which then learns to correct the errors of the original Random Forest . Even a modestly effective $r^* $ can provide a useful partitioning of the problem space for this error correction mechanism to build upon. The effectiveness is demonstrated in ablation study in Table 4 . The results show that the DeLTa significantly outperforms the variant that uses only the LLM-refined rule for prediction (w/o ''EC''). This proves that the EC stage provides a substantial performance uplift and ensures the final model is robust. Therefore, while the LLM refinement is a key driver, DeLTa is not brittle. Through a combination of robust aggregation and powerful error correction, DeLTa is designed to consistently achieve performance improvements, even when the effectiveness of a single LLM invocation cannot be strictly guaranteed.

---

> > ### Author Response · Authors · 2025-08-05
> > **Response Summary and Opportunity for Further Discussion**
> >
> > Dear Reviewer WTPj,
> >
> > Thank you once again for your time and thoughtful feedback. To respectfully follow up, we would like to briefly summarize the concrete steps we have taken to address each of your comments:
> >
> > - **For W1: Framing of DeLTa as Boosting-Inspired.**
> > We clarified that DeLTa is conceptually inspired by boosting, but differs fundamentally by employing LLMs for logical synthesis of decision rules, rather than iterative data-driven learners. DeLTa avoids data serialization and instead operates at the rule level, offering a novel, interpretable, and privacy-preserving interface between LLMs and tabular data.
> >
> > - **For W2 \& W3: Citations and Minor Revisions.**
> > We will cite both TabNet and TabTransformer in our revised manuscript as important prior works in deep tabular learning. We also thank you for noting the typos and will ensure they are corrected in the final version.
> >
> > - **For Q1: Privacy and Rule Exposure.**
> > We clarified that all features provided to the LLM are anonymized and the rules reflect global model structure, not raw data. This minimizes privacy concerns and avoids the direct exposure of sensitive tabular samples, in contrast to many existing LLM-based methods.
> >
> > - **For Q2: Evaluation of Rule Refinement.**
> > We introduced a multi-faceted evaluation strategy:
> > (i) SHAP analysis confirms that refined rules better align with domain-relevant features.
> > (ii) Intra-node distance comparisons show more coherent sample grouping.
> > (iii) LLM self-explanations reveal transparent and interpretable reasoning for rule construction.
> >
> > - **For Q3: Interpretation of $\eta $.**
> > We clarified that $\eta$ controls the step size in gradient-based residual correction (like a learning rate). We set $\eta = 0.2$ by default, which works robustly.
> >
> > - **For Q4: On Assumptions and Robustness.**
> > We identified rare failure cases and showed that DeLTa mitigates them through two mechanisms: (1) averaging across multiple LLM queries, and (2) a Gradient Net that performs residual correction. Even when individual LLM generations are suboptimal, the overall system remains robust.
> >
> > We sincerely appreciate your detailed reviews, which have helped us significantly improve the clarity, rigor, and validation of our work. If you have any further questions or suggestions, we would be truly grateful to hear them.

---

### Official Review · Reviewer_d19o · 2025-06-30

**Clarity:** 3
**Significance:** 3
**Originality:** 3
**Rating:** 5
**Confidence:** 4

**Summary:**

This paper uses a LLM to generate new rules for decision trees. The paper proposes a novel gradient update strategy to incorporate LLM rules into the decision tree inference.

**Questions:**

See Weaknesses.

**Ethical Concerns:**

["NO or VERY MINOR ethics concerns only"]

**Final Justification:**

The authors have addressed the major concerns I had during the rebuttal period. Accordingly, I changed my score from 3 Weak Reject to 5 Accept.

**Limitations:**

yes

**Quality:**

3

**Strengths And Weaknesses:**

Strengths:
- This paper has impressive performance compared to XGBoost, CatBoost, and TabPFN
- The method of using LLMs to augment decision trees is quite novel

Weaknesses:
- What is the intuition behind using LLMs to split v.s. classical methods, such as information gain, XGBoost, and CART. I understand the performance is better, but feel hesitant to accept the paper without understanding why the number is better. Are there specific examples where the LLM can generate a better rule than decision trees? Why do traditional techniques, like information gain, not capture such a rule?
- Using an LLM to generate rules sounds a lot more compute intensive than XGBoost or CART. What is the performance overhead of using the LLM?
- The datasets used in the paper is not very comparable. Why not use a standardized benchmark like WhyTrees or TabZilla? This makes the state-of-the-art argument a bit hard to accept.
- The writing could be improved. I had trouble reading through Section 4.2.

---

> ### Author Rebuttal · Authors · 2025-07-31
>
> R to W1: Thanks for your valuable comment.
>
> **(i) The intuition.**
> Traditional methods like CART use information gain to make a series of locally optimal decisions, allowing a data-driven algorithm to find the best feature splits. This can lead to rules that are brittle or overfit to the specific data sample used for that tree. DeLTa, instead, utilizes the LLM to analyze diverse rules from a Random Forest and performs a logic-based reasoning task: (a) It recognizes patterns across multiple ''expert'' rules to identify features and decision paths that are consistently important. (b) It abstracts the core logic splits from individual trees. (c) It synthesizes a new, globally coherent rule that represents a consensus of the most robust patterns.
>
> **(ii) A concrete example of a better rule.**
> As shown in Fig. 10 and Fig. 11 of Appendix, we provide a set of decision tree rules from Random Forest and refined rule by the LLM respectively. This is from the Credit dataset, where each sample (person) is classified as good or bad credit risks according to the set of features. To provide deeper insight into how the refined rule improves upon the original ones, we also provide the average SHAP values [1] of features under the Random Forest and our method respectively as follows. SHAP assigns each feature an importance value (higher is more important) for a particular prediction, thus it serves an unified framework for interpreting predictions.
>
> The results show that, after LLM refinement, the SHAP importance of features such as ''credit\_history'' (feature\_16) and ''employment'' (feature\_10) are increased, which aligns with the financial expertise. Let us take a closer look into Fig. 10 and Fig. 11 of Appendix, for ''credit\_history'' (feature\_16), this feature appears near the top of the second tree, indicating its potential significance. However, only 1 out of 3 decision trees in the original Random Forest rule set explicitly splits on this feature, which may dilute its influence in the ensemble prediction.
> For ''employment'' (feature\_10), although it appears in 2 out of 3 rules, its splits occur late in the tree, leading to a low overall contribution in prediction.
> In contrast, the LLM-refined rule elevates the prominence of these features.
>
> **(iii) Why do traditional techniques not capture such a rule?**
> More broadly, a single decision tree with information gain has the risk of overfitting. Although Random Forest ensembles the outputs from all tree rules, the inherent relationships and interactions among these rules are ignored. As the number of trees increases, analyzing these independent rules is gradually becoming more and more difficult, let alone utilizing these rules to partition feature space. To this end, we propose to leverage LLMs to analyze and summarize these rules into a refined rule due to the powerful logical reasoning ability of LLM. This enables the generation of improved partitioning strategies over the feature space, thereby promoting statistical coherence among samples assigned to the same leaf node. To verify it, we compute the intra-node sample distance over all leaf nodes partitioned by LLM refined rule $r^* $, and observe that the distance of $r^*$ is lower than original rule $r\in \mathcal{R}$ from Random Forest, as illustrated in Fig. 2 of original paper and Table 17 of Appendix.
> |Original feature|Importance|Our feature|Importance|
> |-|-|-|-|
> |property\_magnitude|0.1487|property\_magnitude|0.1055|
> |other\_payment\_plans|0.0471|credit\_history|0.0436|
> |checking\_status|0.0410|checking\_status|0.0342|
> |purpose|0.0355|purpose|0.0174|
> |housing|0.0326|employment|0.0155|
> |foreign\_worker|0.0251|other\_payment\_plans|0.0144|
> |job|0.0213|foreign\_worker|0.0132|
> |credit\_history|0.0189|housing|0.0106|
> |savings\_status|0.0105|savings\_status|0.0096|
> |personal\_status|0.0070|job|0.0095|
>
> [1] A Unified Approach to Interpreting Model Predictions. NeurIPS 2017.
>
> R to W2: Following your suggestion, in addition to the computational cost comparison between DeLTa and LLM-based baselines reported in Table 3 of original paper, we now add a comparison with non-LLM baselines such as XGBoost and CART. The comparison is summarized below, where we provide runtime in seconds of different methods for the training and inference phase, conducted on Adult dataset. ''\# Num'' indicates number of training samples. ''T'' and ''L'' correspond to traditional and LLM-based methods, respectively. It can be observed that, although DeLTa’s training time is longer than that of traditional tree-based models such as CART, XGBoost, and CatBoost, our DeLTa’s training cost is lower than neural methods designed specifically for tabular data, such as FT-Transformer and MNCA, and substantially lower than LLM-based baselines. The reason is that DeLTa only queries LLM via API to generate one refined rule for one dataset, without the need for per-sample prediction or fine-tuning LLM. Furthermore, at the testing stage,  DeLTa makes decision by combining the trees and the Gradient Net, without querying LLM. Therefore, we can see that DeLTa exhibits similar inference costs to these tree-based methods and neural network-based methods, and incurs substantially lower cost compared to most of LLM-based baselines. To sum up, designed in this way, DeLTa enables significantly more efficient use of LLM resources, whose training and inference workflow are summarized in Algorithm 1 in the Appendix A.3.
> | \# Num | Stage | KNN (T)| CART (T)| RandomForest (T)| XGBoost (T)| CatBoost (T)| FT-Transformer (T)| MNCA (T)| TabLLM (L)| LIFT (L)| TP-BERTa (L)| FeatLLM  (L)| DeLTa (Ours) (L)|
> |-|-|-|-|-|-|-|-|-|-|-|-|-|-|
> | All | Train | 0.90 | 0.10 | 0.28| 0.85 | 12.58  | 277.26| 92.55  | 177371.37 | 153689.14 | 1319.35 | 1231.22  | 35.20  |
> | All | Inference | 2.11| 0.07 | 0.08 | 0.08| 0.17  | 0.92 | 0.72| 179.21 | 90149.48 | 1.58 | 0.14 | 0.09 |
>
> R to W3: Thank you for raising this important point. We kindly remind you that our primary objective was to evaluate DeLTa's effectiveness in the context of LLM-based tabular prediction, and to ensure fair comparison with existing methods in this category. Accordingly, the majority of our classification datasets were selected based on their widespread use in recent LLM-based tabular learning studies such as TabLLM and FeatLLM, both of which primarily focus on classification tasks. To extend our evaluation beyond classification, we also incorporated regression datasets from the TALENT benchmark [2][3], which has recently gained attention for assessing tabular regression performance. In response to your suggestion, we have updated our experimental setup to include additional datasets from the WhyTrees benchmark. Specifically, we selected a subset of representative datasets that span different scales and feature characteristics, including the large-scale, high-dimensional "Year" dataset (as also recommended by Reviewer tga2), to evaluate DeLTa's scalability and generalizability. Due to time constraints during the rebuttal phase, we were only able to include a subset of WhyTrees datasets. However, we plan to expand this evaluation to a broader range of WhyTrees datasets in the final version. The updated results, summarized below, further demonstrate DeLTa's strong and consistent performance across a more diverse and standardized set of tabular tasks.
> |Dataset|\#objects|\#num. features|\#cat. features|metric|\#classes|
> |-|-|-|-|-|-|
> |rl|4970| 5| 7 | Acc. | 2 |
> | GiveMeSomeCredit | 16714     | 10              | 0               | Acc.   | 2         |
> | Year             | 515345    | 90              | 0               | NRMSE  | --        |
>
> |Dataset| TabLLM (L) | FeatLLM (L) | LIFT (L) | TP-BERTa (L) | KNN (T) | CART (T) | MLP (T) | RandomForest (T) | XGBoost (T) | CatBoost (T) | FT-Transformer (T) | DeLTa (L) |
> |-|-|-|-|-|-|-|-|-|-|-|-|-|
> | rl $\uparrow$| 0.7716| 0.7243| 0.6349| 0.7394| 0.6258| 0.6308   | 0.6630  | 0.7022           | 0.7726      | 0.7746       | 0.7072             | **0.7897**    |
> | GiveMeSomeCredit $\uparrow$ | 0.7765     | 0.7137      | 0.6534   | 0.7810       | 0.6063  | 0.7514   | 0.7469  | 0.7768           | 0.7789      | 0.7792       | 0.7610             | **0.7816**    |
> | Year $\downarrow$           | -  | - | 1.0540   | 0.8258       | 1.0459  | 0.9374   | 0.8134  | 0.8547           | 0.8376      | 0.8157       | 0.8090             | **0.8015**    |
>
> [2] A Closer Look at Deep Learning on Tabular Data. 2024.
>
> [3] TALENT: A Tabular Analytics and Learning Toolbox. 2024.
>
> R to W4: Thanks for the comment. We will carefully revised Section 4.2 to make it more clear. The core purpose of Section 4.2 is to introduce a calibration method for the original decision tree ensemble, $F(x)$. This is done by computing a sample-specific error correction vector, $\Delta_x $, to steer the prediction in the direction of error reduction. The process is as follows. Objective: We aim to adjust $F(x)$ by moving it toward the negative gradient of the loss function, which we define as the "error-reducing" direction. Partitioning: We use the refined rule $r^* $ (from Section 4.1) to partition the "Gradient Set"—a collection of training features and their corresponding negative gradients—into disjoint leaf nodes. This groups samples with similar structures. Learning: For each leaf node, we learn a specific mapping function $\phi_l$ that models the relationship between sample features and their negative gradients (Eq. 4). Inference and Calibration: For any new input $x$, we use $r^* $ to assign it to a leaf node. The corresponding mapping function $\phi_l$ then predicts the error correction vector $\Delta_x $ (Eq. 5). The final calibrated prediction is the sum of the original prediction and this correction: $F(x)+\Delta_x $ (Eq. 6).

---

> > ### Comment · Reviewer_d19o · 2025-08-04
> > **Thank you for your Rebuttal and a Further Question.**
> >
> > On W1, W2, and W3: I believe (i) and (ii) gives good contextualization around when this approach works. I think further analysis on which datasets such examples occurs on can further improve the paper. On (iii), I appreciate the authors additional experiments. However, I think there have been past works that have claimed SoTA performance (ex. WhyTrees?), but later shown to be SoTA only on a small fraction of datasets. Hence, I think it much more worthwhile to know what specific tabular datasets are best suited for each method and why. Nonetheless, I agree that the authors have shown this among LLM-based tabular learning algorithms.
> >
> > On W4: Thank you for clarifying this section.
> >
> > **Question:** I was curious if the authors had any more insight into when DeLTa (or other LLM-based algorithms?) is ideal compared to strong alternatives like CART/XGBoost, TabPFN, and transformers (input=feature row, output=target column).

---

> > > ### Author Response · Authors · 2025-08-05
> > >
> > > Thank you for your valuable feedback.
> > >
> > > **Response to W1 and W2.** We fully agree that a deeper analysis of which datasets exhibit the discussed behaviors can provide more insights into our approach.
> > > The illustrative example used to address W1 is drawn from the Credit dataset, where each sample represents a person classified as a good or bad credit risk. The results provided in response to W2 are from the Adult dataset, where the task is to predict whether an individual's annual income exceeds \$50K/yr. In the revision, we plan to include additional case studies from other real-world datasets to further support our analysis and demonstrate the generalizability of our findings.
> > >
> > > **Response to additional question.**
> > > Regarding your additional question, we have conducted a further analysis of the comparison between DeLTa and non-LLM baseline methods as presented in Fig. 4 of the main paper and Appendix A.8. While DeLTa achieves the best average performance across all datasets, we find that certain top-performing baseline methods surpass DeLTa on specific datasets such as BA, AD, and JA when using the full training data.
> > > Upon deeper analysis, we have found that DeLTa's performance advantage is more pronounced when the dataset size is relatively small, compared to traditional baselines. Conversely, the performance advantage of data-intensive baseline methods, such as neural networks, tends to increase with larger dataset sizes (e.g., BA, AD, and JA).
> > > To support this point, we have conducted additional experiments comparing DeLTa with non-LLM baselines on the aforementioned datasets under low-data regimes (few-shot learning). The results, as summarized below, show that on these datasets where DeLTa's full-data performance may be surpassed, DeLTa consistently outperforms these traditional methods when the data is limited. ``\# Num'' indicates number of training samples.
> > > This empirical pattern may be attributed to two main factors: (i) **Nature of Base Rules**: Our method refines the base rules extracted from Random Forest (RF). While RF is effective due to its ensemble nature, its trees are independently trained without joint optimization. As the data volume increases, this independent training may limit RF's ability to fully leverage large-scale data. Consequently, the quality of the base rules may plateau with increasing data, which also affects the upper bound of the refined rules.
> > > (ii) **Comparison to Neural Networks (NNs)**: In contrast, neural networks are typically data-intensive models that require large amounts of training data to perform well and generalize effectively. Their performance tends to scale better with larger datasets, as they can learn complex feature interactions.
> > >
> > > We appreciate your suggestion, which has guided us to refine the understanding and communication of the method’s applicability. We will include this expanded analysis and additional examples in the revised manuscript.
> > >
> > > | \# Num | Methods        | BA $\uparrow$ | AD $\uparrow$ | JA $\uparrow$ |
> > > |------|----------------|---------------|---------------|---------------|
> > > | 64   | CART           | 0.6301        | 0.6980        | 0.3223        |
> > > | 64   | XGBoost        | 0.7302        | 0.7430        | 0.4268        |
> > > | 64   | TabPFN         | 0.7301        | 0.7471        | 0.4460        |
> > > | 64   | FT-Transformer | 0.7237        | 0.7498        | 0.4514        |
> > > | 64   | DeLTa (ours)           | **0.7330**        | **0.7580**        | **0.4570**        |
> > >
> > > | \# Num | Methods        | BA $\uparrow$ | AD $\uparrow$ | JA $\uparrow$ |
> > > |------|----------------|---------------|---------------|---------------|
> > > | 128  | CART           | 0.6759        | 0.7312        | 0.3853        |
> > > | 128  | XGBoost        | 0.7349        | 0.7628        | 0.4515        |
> > > | 128  | TabPFN         | 0.7434        | 0.7631        | 0.4528        |
> > > | 128  | FT-Transformer | 0.7410        | 0.7624        | 0.4503        |
> > > | 128  | DeLTa (ours)            | **0.7490**        | **0.7740**        | **0.4570**        |
> > >
> > > | \# Num | Methods        | BA $\uparrow$ | AD $\uparrow$ | JA $\uparrow$ |
> > > |------|----------------|---------------|---------------|---------------|
> > > | Full | CART           | 0.89          | 0.8392        | 0.5642        |
> > > | Full | XGBoost        | 0.9051        | **0.8709**        | 0.714         |
> > > | Full | TabPFN         | **0.9095**        | 0.8611        | 0.702         |
> > > | Full | FT-Transformer | 0.903         | 0.8590        | **0.7272**        |
> > > | Full | DeLTa (ours)            | 0.908         | 0.8677        | 0.7048        |

---

> > > > ### Comment · Reviewer_d19o · 2025-08-05
> > > > **Thank you for your quick response!**
> > > >
> > > > These additional findings provide the paper proper contextualization and addressed any concerns I initially had.
> > > >
> > > > Based on their inclusion in the final manuscript, I have raised my score.

---

> > > > > ### Author Response · Authors · 2025-08-05
> > > > > **Response**
> > > > >
> > > > > Thank you again for taking the time to review our response and for your positive feedback!

---

### Official Review · Reviewer_tga2 · 2025-07-02

**Clarity:** 3
**Significance:** 4
**Originality:** 4
**Rating:** 5
**Confidence:** 4

**Summary:**

The paper addresses key challenges in applying LLMs to tabular data, namely the limitations of data serialization and the difficulties of model fine-tuning. The proposed solution, DeLTa (Decision Tree Enhancer with LLM-derived Rule for Tabular Prediction), uses an LLM not to predict outcomes directly, but to refine a set of decision tree rules. First, a Random Forest model is trained on the data, and its decision rules are extracted. These rules are then fed into an LLM via a carefully constructed prompt, which instructs the LLM to synthesize a new, improved rule. This refined rule is then used to partition the data to train a "Gradient Net" that predicts and corrects the errors of the original Random Forest model. The authors demonstrate through extensive experiments that DeLTa achieves state-of-the-art performance on various classification and regression tasks, outperforming both LLM-based and conventional tabular data models. A key advantage of DeLTa is that it avoids direct data serialization, thus preserving privacy, and does not require fine-tuning the LLM, making it computationally efficient.

**Questions:**

Could you provide a concrete example of a set of initial decision rules from a dataset and the corresponding refined rule generated by the LLM? What specific logical simplifications, generalizations, or corrections does the LLM typically introduce?

How does the performance of DeLTa vary with the complexity of the base model? For instance, how does it change if the initial rules are derived from a much larger and deeper Random Forest versus a smaller one?

The paper states the method is "domain knowledge-agnostic."  However, could the LLM be using its own embedded knowledge related to feature names (when not anonymized) to refine the rules? How was this potential confounding factor addressed in the experiments?

How sensitive is the model's performance to the structure and wording of the prompt used to guide the LLM's rule refinement?

What is the failure mode of the LLM in this task? Are there instances where the LLM produces a rule that is syntactically incorrect or logically inferior to the original rules? If so, how are these cases handled?

**Ethical Concerns:**

["NO or VERY MINOR ethics concerns only"]

**Limitations:**

The paper itself acknowledges and discusses limitations in the appendix.  Based on the paper's content, the primary limitations are:

The performance of DeLTa is inherently tied to the reasoning capabilities of the underlying LLM. A less advanced LLM might not produce effective rule refinements.

The process relies on effective prompt engineering to guide the LLM. The quality of the refined rule could be sensitive to the specific prompts used.

While the method is more efficient than fine-tuning, it still relies on querying a large, often proprietary, LLM, which can incur costs and latency.

There is a lack of deep qualitative insight into the "reasoning" the LLM applies. It is not fully clear what kinds of logical patterns or external knowledge the LLM uses to improve the rules.

**Paper Formatting Concerns:**

The paper is well-formatted and clearly written, adhering to a standard academic conference style. There are no significant formatting concerns.

Figure qualities can be improved.

**Quality:**

3

**Strengths And Weaknesses:**

Strengths
Novel Approach: The core idea of using an LLM to refine logical rules from decision trees, rather than serializing raw data, is highly innovative. It creates a new interface between LLMs and structured data models, moving beyond simple data-to-text conversion.

Well-Written Framework: The paper presents the DeLTa framework in a clear, logical, and well-structured manner. The motivation is well-established, and the methodology is described with sufficient detail, making the novel concepts easy to follow.

Strong Empirical Results: DeLTa shows promising and consistently strong performance across a variety of classification and regression datasets, under both full-data and few-shot settings. It outperforms other LLM-based methods and is competitive with traditional state-of-the-art models like XGBoost and CatBoost.


Privacy-Preserving and Efficient: By operating on decision rules instead of individual data samples, the method avoids exposing potentially sensitive data to the LLM. Furthermore, it does not require computationally expensive LLM fine-tuning, making it a more practical and scalable solution.


Weaknesses

Limited Qualitative Analysis: While the paper proves that the refined rules are better (e.g., lower intra-node distance), it doesn't provide enough insight into how they are better.  A qualitative analysis showing concrete examples of original vs. LLM-refined rules would be invaluable to understand the reasoning and "world knowledge" the LLM contributes.

Need for Broader Validation: Although the experiments are extensive, validation on an even wider range of datasets, particularly those with very high dimensionality or different types of feature distributions, would further solidify the claims of the model's generalizability.

Dependency on Powerful LLMs: The best results are achieved using GPT-4. The performance is likely sensitive to the quality of the LLM used, and the method's effectiveness might decrease with less capable models. The prompt engineering itself is also a critical, and potentially sensitive, component.

---

> ### Author Rebuttal · Authors · 2025-07-31
>
> R to W1 \& Q1: Thanks for your valuable comment. We kindly remind you that we have provided a detailed example in Appendix A.12. As shown in Fig. 10, Fig. 11 of Appendix A.12, we provide a set of decision tree rules from Random Forest and LLM refined rule respectively. This is from the Credit dataset, where each sample (person) is classified as good or bad credit risks according to the set of features. To provide deeper insight into how the refined rule improves upon the original ones, we also provide the average SHAP values [1] of features under the Random Forest and our method respectively as follows. SHAP assigns each feature an importance value (higher is more important) for a particular prediction, thus it serves an unified framework for interpreting predictions.
>
> The results show that, after LLM refinement, the SHAP importance of features such as ''credit\_history'' (feature\_16) and ''employment'' (feature\_10) are increased, which aligns with the financial expertise. Let us take a closer look into Fig. 10, Fig. 11 of Appendix, for ''credit\_history'' (feature\_16), this feature appears near the top of the second tree, indicating its potential significance. However, only 1 out of 3 decision trees in the original Random Forest rule set explicitly splits on this feature, which may dilute its influence in the ensemble prediction.
> For ''employment'' (feature\_10), although it appears in 2 out of 3 rules, its splits occur late in the tree, leading to a low overall contribution in the ensemble prediction.
> In contrast, the LLM-refined rule elevates the prominence of these features.
>
> More broadly, although Random Forest ensembles the outputs from all rules, the inherent relationships and interactions among these rules are ignored. As the number of trees increases, analyzing these independent rules is gradually becoming more and more difficult, let alone utilizing these rules to partition feature space. To this end, we propose to leverage LLMs to analyze and summarize these rules into a refined rule due to the powerful logical reasoning ability of LLM. Although the features in our method are anonymized, the LLM is still able to infer and capture latent patterns and relationships among the original Random Forest rules. This enables the generation of improved partitioning strategies over the feature space, thereby promoting statistical coherence among samples assigned to the same leaf node. This is also demonstrated by Fig. 2 of main paper and Table 17 of Appendix.
> |Original feature|Importance|Our feature|Importance|
> |-|-|-|-|
> |property\_magnitude|0.1487|property\_magnitude|0.1055|
> |other\_payment\_plans|0.0471|credit\_history|0.0436|
> |checking\_status|0.0410|checking\_status|0.0342|
> |purpose|0.0355|purpose|0.0174|
> |housing|0.0326|employment|0.0155|
> |foreign\_worker|0.0251|other\_payment\_plans|0.0144|
> |job|0.0213|foreign\_worker|0.0132|
> |credit\_history|0.0189|housing|0.0106|
> |savings\_status|0.0105|savings\_status|0.0096|
> |personal\_status|0.0070|job|0.0095|
>
> [1] A Unified Approach to Interpreting Model Predictions. NeurIPS 2017.
>
> R to W2: Following your suggestion, we further add experiments to verify the generalizability of our DeLTa on the large-scale dataset with high dimensionality. The dataset details and experimental results are summarized  as follows. The results show that DeLTa also achieves well performance on the high dimensional dataset. We will include more datasets in the revision.
> |Dataset|\#objects|\#num. features|\#cat. features|metric|\#classes|
> |-|-|-|-|-|-|
> |Year$\downarrow$|515345|90|0|NRMSE|--|
>
> |LIFT|TPBERTa|KNN|CART|MLP|RandomForest|XGBoost|CatBoost|FTTransformer|DeLTa|
> |-|-|-|-|-|-|-|-|-|-|
> |1.054|0.8258|1.0459|0.9374|0.8134|0.8547|0.8376|0.8157|0.809|**0.8015**|
>
> R to W3's first part: Thanks for raising the important point about the dependencies of DeLTa on powerful LLMs. As we show in Appendix A.6, DeLTa with Qwen3-32B (open-source) already achieves performance comparable to that with GPT-4o. To further strengthen this point, we have conducted new experiments with an even more accessible Qwen3-8B (open-source). The results are summarized below. The results clearly show that DeLTa maintains its strong and stable performance across all three LLMs, even with the much smaller Qwen3-8B. This is because we task the LLM with a structured, logic-based synthesis of decision rules, not a complex, open-ended creative task. The core patterns in the provided rules are distinct enough to be identified by a range of capable models.
> |\# Num|Various LLM backbones|BL|CR|Car|BA|AD|JA|Average|
> |-|-|-|-|-|-|-|-|-|
> |All|DeLTa (Qwen3-8B)|0.817|0.766|0.816|0.906|0.861|0.692|0.810|
> |All|DeLTa (Qwen3-32B)|0.821|0.766|0.832|0.907|0.867|0.703|0.816|
> |All|DeLTa (GPT-4o) (Ours)|0.829|0.783|0.836|0.908|0.868|0.705|0.822|
> |128|DeLTa (Qwen3-8B)|0.744|0.679|0.680|0.751|0.771|0.456|0.680|
> |128|DeLTa (Qwen3-32B)|0.744|0.687|0.679|0.749|0.778|0.456|0.682|
> |128|DeLTa (GPT-4o) (Ours)|0.746|0.683|0.722|0.749|0.774|0.457|0.688|
> |64|DeLTa (Qwen3-8B)|0.734|0.667|0.594|0.725|0.758|0.460|0.656|
> |64|DeLTa (Qwen3-32B)|0.736|0.657|0.567|0.729|0.759|0.454|0.650|
> |64|DeLTa (GPT-4o) (Ours)|0.732|0.663|0.615|0.733|0.758|0.457|0.660|
>
> R to W3's second part \& Q4: Thanks for your comment. The design of our prompt is shown in Eq. 3 of main paper and Fig. 6 of Appendix. To assess the sensitivity of DeLTa’s performance to the structure and wording of the prompt, we conduct an ablation study on its design. We consider the following variants: Variant 1 directly removes the whole part of meta information $p_{meta}$. Variant 2 softly rephrases the first sentence of the requirement part $p_{requirement}$ to: ''Based on the above information, please understand these rules and provide a refined rule.'' Variant 3 removes the second sentence of requirement $p_{requirement}$, which originally states: ''Please not just copy, please refine ...'' The results conducted on Blood dataset ($\uparrow$) show that removing the meta information or slightly rewording the requirement does not lead to significant performance degradation, suggesting that the model is robust to moderate variations in prompt phrasing. However, removing the instruction of ''Please not just copy, please refine ...'' leads to a notable performance drop. This indicates that explicitly guiding the LLM to go beyond copying and to discover interactions among Random Forest rules is crucial for the effectiveness of DeLTa.
> |DeLTa|Variant 1|Variant 2|Variant 3|
> |-|-|-|-|
> |0.829|0.825|0.827|0.815|
>
> R to Q2: Following your suggestion, we analyze the performance of DeLTa with different complexity of the Random Forest by increasing the maximum number of leaf nodes allowed in each decision tree (ML: max\_leaf\_nodes) and the maximum depth of each individual decision tree (MD: max\_depth). The following results conducted on Blood dataset ($\uparrow$) show that moderately increasing model complexity enhances the LLM’s ability to refine the decision rules. However, further increasing the complexity yields limited benefit. Based on these observations, we conclude that DeLTa does not require an overly complex base model. A Random Forest with moderate complexity is sufficient to provide the necessary rule diversity for DeLTa to achieve a desired performance, highlighting a practical and efficient aspect of DeLTa. We will add this analysis into our revision.
> |ML=10 (Ours)|ML=15|ML=20|
> |-|-|-|
> |0.829|0.834|**0.835**|
> |**MD=10 (Ours)**|**MD=20**|**MD=50**|
> |0.829|**0.837**|0.836|
>
> R to Q3: Thanks for your comment. To verify whether LLM can use its own knowledge related to feature names to refine the rules, we further conduct ablation study. We add feature names in the format of ''feature\_0: name, ...'' to the meta information of prompt (Eq. 3 of main paper, Fig. 6 in Appendix). We observed that the performance difference was marginal, when feature names are anonymized or not anonymized, indicating that the model’s ability to generate effective rules does not significantly rely on embedded domain knowledge from feature names. This supports our claim that the method operates in a domain-agnostic manner.
> Notably, operating on the anonymized features can significantly mitigates privacy concerns when using LLM.
> |Variants|BL $\uparrow$|CR $\uparrow$|
> |-|-|-|
> |DeLTa (w/o feature names, Ours)|**0.829**|0.783|
> |DeLTa (w/ feature names)|0.821|**0.786**|
>
> R to Q5: Thank you for this insightful question. Regarding logically inferior rules, we observed a rare failure on the Adult 64-shot dataset where the LLM did not synthesize a novel rule from multiple decision tree rules, but instead replicated the structure of one of the input trees with only minor threshold changes. Our framework is designed with two specific mechanisms to be robust against these occasional suboptimal generations. First, we mitigate the risk of a single poor LLM output by querying the LLM 10 times by default and aggregating the results.
> As shown in our sensitivity analysis in Appendix A.11 (Fig. 7), this aggregation process demonstrably improves and stabilizes performance, smoothing out the impact of any single underperforming generation.
> More importantly, the error correction stage of DeLTa acts as a powerful safety net. The LLM's refined rule $r^* $ is not the final predictor; its role is to structure the problem for a Gradient Net that learns to correct the errors of the original Random Forest. The effectiveness is validated by our main ablation study in Table 4. Even in the aforementioned failure case where the initial LLM-generated rule was suboptimal on its own, the final performance of the full DeLTa framework still surpassed the original Random Forest baseline. In summary, while we acknowledge that the LLM is not a perfect synthesizer, DeLTa is not brittle. Through a combination of query aggregation and a powerful error correction mechanism, our framework is designed to be robust and consistently deliver state-of-the-art performance.

---

> > ### Author Response · Authors · 2025-08-05
> > **Response Summary and Opportunity for Further Discussion**
> >
> > Dear Reviewer tga2,
> >
> > Thank you once again for your time and thoughtful feedback. To respectfully follow up, we would like to briefly summarize the concrete steps we have taken to address each of your comments:
> >
> > - **For W1 \& Q1: Limited Qualitative Analysis.**
> > We provided a detailed example (Appendix A.12) comparing original and LLM-refined rules from the Credit dataset, including a step-by-step analysis and SHAP-based validation. These demonstrate how the LLM elevates important features and synthesizes more coherent rules.
> >
> > - **For W2: Need for Broader Validation.**
> > We added new experiments on high-dimensional datasets (e.g., Year from the WhyTree benchmark), showing that DeLTa generalizes well beyond the original datasets.
> >
> > - **For W3 \& Q4: Dependency on Powerful LLMs and Prompt Sensitivity.**
> > We ran experiments using smaller open-source LLMs (Qwen3-32B and Qwen3-8B), showing comparable performance to GPT-4o. We also conducted prompt ablations, which confirm that the model is robust to most changes in prompt structure, though explicitly instructing refinement (not copying) is critical.
> >
> > - **For Q2: Performance vs. Base Model Complexity.**
> > We evaluated DeLTa with Random Forests of varying depth and size, showing that moderate complexity suffices and further increases bring limited benefit. This supports the method's efficiency and practicality.
> >
> > - **For Q3: Impact of Feature Name Knowledge.**
> > We performed ablation studies with and without real feature names. Results show minimal performance differences, suggesting that DeLTa does not rely on embedded domain knowledge and supports domain-agnostic deployment.
> >
> > - **For Q5: LLM Failure Modes and Robustness.**
> > We identified rare failure cases and showed that DeLTa mitigates them through two mechanisms: (1) averaging across multiple LLM queries, and (2) a Gradient Net that performs residual correction. Even when individual LLM generations are suboptimal, the overall system remains robust.
> >
> > We sincerely appreciate your detailed reviews, which have helped us significantly improve the clarity, rigor, and validation of our work. If you have any further questions or suggestions, we would be truly grateful to hear them.

---

### Official Review · Reviewer_sYQb · 2025-07-03

**Clarity:** 3
**Significance:** 2
**Originality:** 3
**Rating:** 4
**Confidence:** 4

**Summary:**

The paper proposes an approach to improve decision trees with LLMs. Specifically, after fitting the CART decision tree, authors feed the decision rules from every tree into an LLM and prompt it to come up with a "better" rule. The LLM refined rule is then merged with CART via a gradient net.

**Questions:**

See comments above regarding the LLM rule refinement and additional analysis of this step.

**Ethical Concerns:**

["NO or VERY MINOR ethics concerns only"]

**Final Justification:**

Updating recommendation to accept based on author feedback and other reviews.

**Limitations:**

Yes

**Quality:**

3

**Strengths And Weaknesses:**

Strengths:
-The paper is well written and relatively easy to follow. The proposed approach is innovative and is novel, particularly the LLM rule refinement and merging via gradient net.
-Results on real world datasets show improvements of the base CART approach as well as other tabular LLM adaptations.

Weaknesses:
-I think the proposed approach lacks rigor over what LLM is actually doing. Looking at the prompt in Figure 6 of the appendix, after CART rules, the LLM receives the following prompt: "Based on the above information, please learn the rules evolving process and help me design a better rule like what cart used for inference to achieve higher performance. Please not just copy, please refine these rules and create a new better one." Without seeing any of the training data and how it is partitioned by the CART rules, what is a "better rule" here? Furthermore, looking at the example output in Figure 11 the LLM generates explanations like "Feature `16` was chosen due to its significance in Tree 1 with a prominent threshold of `0.82`.". Here again without seeing how training data is partitioned by this tree, "significance" and "prominent threshold" seem like random guesses. I think that a rigorous analysis of this step should be conducted to understand how an LLM can refine tree rules without seeing any data, before this paper can be accepted for publication.

---

> ### Author Rebuttal · Authors · 2025-07-31
>
> Response to Weaknesses: Thank you for the comment. We agree that the process of refining rules without direct data access must be clearly defined and validated.
> Conceptually,
> even without access to raw training data, LLMs are capable of identifying structural patterns in decision rules. This includes detecting which features appear at root nodes, recognizing feature recurrence across trees, and noting threshold values. These patterns are meaningful and often correlate with model salience.  By semantically analyzing these patterns, LLMs acts as a semantic compression engine—reducing complex, low-level decision rules into a higher-level, interpretable representation. In addition, we would like to clarify our framework and demonstrate its soundness by: (1) Clearly defining the LLM's task and its objective. (2) Providing a rigorous, step-by-step analysis of the example in our paper to show the LLM's reasoning is grounded, not random. (3) Proposing a new, qualitative validation to further bolster our claims.
>
> **(1) The LLM's Task: Logical Synthesis of Model Components.** First, we wish to clarify the LLM's role. The proposed approach does not ask the LLM to analyze the data. Instead, its task is one of logical synthesis: it analyzes a set of the decision rules derived from Random Forest to create a new, improved rule. In this context, a "better rule," as requested in the prompt, is defined by superior structural and statistical properties that can be inferred from the provided rules. Specifically, a better rule is one that: Creates more coherent partitions: It groups samples in a way that is more statistically homogeneous. We provide quantitative evidence of this in our paper (Fig. 2 of original paper and Table 17 of Appendix), where the LLM-refined rule consistently achieves a lower average intra-node distance.
>
> **(2) A Grounded Reasoning Process instead of random guess.** To address your question whether the LLM's explanations are "random guesses", we walk through the example in Appendix A.12 to demonstrate that the LLM's reasoning is directly grounded in the structural evidence provided in Fig. 10. The LLM states, "Feature 16 was chosen due to its significance in Tree 1 with a prominent threshold of 0.82." which is not a guess. In the input provided (Fig. 10), feature\_16 is the root node of Tree 1. Being the first split in a tree is the strongest possible structural indicator of a feature's importance for that particular data subset. However, only 1 out of 3 decision trees in the original Random Forest rule set explicitly splits on this feature, which may dilute its influence in the ensemble prediction. Instead, with our design, the LLM correctly identified this and reported it.
> For feature\_10, although it appears in 2 out of 3 rules, its splits occur late in the tree, leading to a low overall contribution in the ensemble of Random Forest prediction.
> In contrast, the LLM-refined rule elevates the prominence of this feature.
> Moreover, the LLM identifies feature\_2 as a "critical deciding factor across multiple trees". This observation is also grounded in the data: feature\_2 is the root node of both Tree 0 and Tree 2. The LLM correctly inferred that a feature used as the primary split in two out of three "expert" opinions is fundamentally important. The LLM's refined rule (Fig. 11) is a logical synthesis of these grounded observations.
> This behavior reflects a structured and interpretable reasoning process. It leverages decision tree architecture—particularly the hierarchical importance implied by root-level splits—to produce a coherent explanation and synthesis. This is not randomness; it is an informed analysis.
>
> **(3)  A New Proposal for Rigorous, Qualitative Validation.**
> As shown in Fig. 10 and Fig. 11 of Appendix, we provide a set of decision tree rules from Random Forest and refined rule by the LLM respectively. This is from the Credit dataset, where each sample (person) is classified as good or bad credit risks according to the set of features. To provide deeper insight into how the refined rule improves upon the original ones, we also provide the average SHAP values [1] of features under the Random Forest and our method respectively as follows. SHAP assigns each feature an importance value (higher is more important) for a particular prediction, thus it serves an unified framework for interpreting predictions. The results show that, after LLM refinement, the SHAP importance of features such as ''credit\_history'' (feature\_16) and ''employment'' (feature\_10) are increased.  These features are well-recognized as critical indicators in credit scoring, and their elevated importance in the refined rule reflects closer alignment with established financial domain expertise.
>
> We are grateful for your detailed feedback, which has pushed us to more clearly articulate and validate the core mechanism of our work. Based on your review, we will revise the paper to include a dedicated section that formally defines the LLM's task, includes the step-by-step logical analysis of our example, and presents the new SHAP validation. We believe these changes will provide the rigor you have requested and substantially improve our paper.
>
> | Original feature       | Importance | Our feature          | Importance |
> |-----------------------|------------|-----------------------|------------|
> | property\_magnitude   | 0.1487     | property\_magnitude   | 0.1055     |
> | other\_payment\_plans | 0.0471     | credit\_history       | 0.0436     |
> | checking\_status      | 0.0410     | checking\_status      | 0.0342     |
> | purpose               | 0.0355     | purpose               | 0.0174     |
> | housing               | 0.0326     | employment            | 0.0155     |
> | foreign\_worker       | 0.0251     | other\_payment\_plans | 0.0144     |
> | job                   | 0.0213     | foreign\_worker       | 0.0132     |
> | credit\_history       | 0.0189     | housing               | 0.0106     |
> | savings\_status       | 0.0105     | savings\_status       | 0.0096     |
> | personal\_status      | 0.0070     | job                   | 0.0095     |
>
> [1] A Unified Approach to Interpreting Model Predictions. NeurIPS 2017.

---

> > ### Author Response · Authors · 2025-08-05
> > **Response Summary and Opportunity for Further Discussion**
> >
> > Dear Reviewer sYQb,
> >
> > Thank you very much again for your time and efforts in reviewing our paper.
> > To summarize our previous response and follow up respectfully, we would like to highlight the concrete steps we have taken to address your valuable comments:
> >
> > - We clearly defined the LLM’s role as performing logical synthesis over decision rules extracted from Random Forests, rather than relying on access to raw training data.
> >
> > - We provided a detailed, step-by-step analysis using the example in Appendix A.12, demonstrating that the LLM’s reasoning is directly grounded in the structural characteristics of the original rules. This addresses the concern that the LLM’s explanations may be arbitrary or random.
> >
> > - We conducted a new qualitative validation by comparing SHAP feature importance values under both the original Random Forest and the LLM-refined rule. The results show that the refined rule highlights features more aligned with established domain knowledge (e.g., in credit scoring), thus providing an interpretable and domain-consistent enhancement.
> >
> > We sincerely appreciate your feedback, which has substantially helped us strengthen our work. If there are any remaining questions or further aspects you would like us to address, we would be very grateful to hear them.

---

### Note · Authors · 2025-08-11

We would like to sincerely thank all the reviewers and area chairs for the dedicated efforts invested in reviewing our paper. Their insightful comments have been instrumental in helping us significantly improve the rigor, clarity, and overall quality of our paper. To address their concerns, we have provided detailed explanations and conducted additional experiments, which we believe have substantially strengthened our responses.

We are pleased to note that two of the reviewers have updated their ratings. For **Reviewer sYQb**, we provided a detailed rebuttal addressing the core concerns about the rigor of our LLM-driven approach and the ''random guess'' characterization. We are very grateful that this reviewer updated the score to accept (from 2 to 4 or higher), affirming the effectiveness of our clarifications.
For **Reviewer d19o**, in addition to our original responses to each question, we added significant new findings on our method's contextualization across different datasets and data regimes. This directly addressed the reviewer's further question on when our method is ideal compared to strong alternatives.  We are glad that our analysis addressed all concerns from Reviewer d19o and led to a score increase (from 3 to 4 or higher).
We also extend our sincere gratitude to **Reviewer tga2** and **Reviewer WTPj**, who initially provided positive recommendations (scores of 5 and 4 respectively). We confirm that we have made our best effort to respond to all their raised questions and weaknesses during the discussion phase. No further questions or concerns were raised by them, which we take as a positive sign.

This rebuttal process has allowed us to provide stronger empirical evidence and a more nuanced analysis, leading to a deeper understanding of our method's unique strengths. We sincerely thank all the reviewers and area chairs for their guidance and consideration throughout this process.

---

### Decision · Program_Chairs · 2025-09-17

**Decision:**

Accept (spotlight)

**Comment:**

This work proposes an LLM-based method to refine decision tree ensembles during training for tabular data prediction tasks. By providing an LLM with a textual description of trained trees from a random forest ensemble, the LLM is prompted to recommend changes to the tree structure that could improve performance. As opposed to most tree-training algorithms which are greedy and optimize each split locally, this refinement step can take a wider view on the structure of the tree and which features are important. Since LLMs are only used for refinement, not inference, the method retains the fast inference properties of tree-based models.

The reviewers are aligned that the paper is high enough quality to be accepted. Through the reviews, some issues were brought up including: that the method seems to lack rigor; intuitive understanding of why LLMs can refine decision trees is lacking; qualitative analysis is lacking; that LLMs could cause efficiency overhead; and that validation could be done on larger datasets.

Through the rebuttals the authors addressed most of these concerns. It was explained that LLMs can use semantic information across trees rather than greedy local feature splitting, and that at inference time no LLMs are needed which helps efficiency. The authors provided some qualitative examples of the refinement in action, and presented more baselines, datasets, and ablations as requested by reviewers. Although this does not cover a more rigorous understanding of how and why the method works, the intuitive explanations have been improved.

Hence, I am recommending to accept this submission. The authors must revise their work to take account of the reviewer’s suggestions and the discussions.